# A neuromorphic model of active vision shows how spatiotemporal encoding in lobula neurons can aid pattern recognition in bees

HaDi MaBouDi[1,2,3,4]*, Mark Roper[4,5], Marie-Geneviève Guiraud[4,6], Mikko Juusola[2,3], Lars Chittka[4], James AR Marshall[1,7]

[1]Department of Computer Science, University of Sheffield, Sheffield, United Kingdom; [2]School of Biosciences, University of Sheffield, Sheffield, United Kingdom; [3]Neuroscience Institute, University of Sheffield, Sheffield, United Kingdom; [4]School of Biological and Behavioural Sciences, Queen Mary University of London, London, United Kingdom; [5]Drone Development Lab, Ben Thorns Ltd, Colchester, United Kingdom; [6]School of Natural Sciences, Macquarie University, North Ryde, Australia; [7]Opteran Technologies Ltd, Sheffield, United Kingdom

**Abstract** Bees' remarkable visual learning abilities make them ideal for studying active information acquisition and representation. Here, we develop a biologically inspired model to examine how flight behaviours during visual scanning shape neural representation in the insect brain, exploring the interplay between scanning behaviour, neural connectivity, and visual encoding efficiency. Incorporating non-associative learning—adaptive changes without reinforcement—and exposing the model to sequential natural images during scanning, we obtain results that closely match neurobiological observations. Active scanning and non-associative learning dynamically shape neural activity, optimising information flow and representation. Lobula neurons, crucial for visual integration, self-organise into orientation-selective cells with sparse, decorrelated responses to orthogonal bar movements. They encode a range of orientations, biased by input speed and contrast, suggesting co-evolution with scanning behaviour to enhance visual representation and support efficient coding. To assess the significance of this spatiotemporal coding, we extend the model with circuitry analogous to the mushroom body, a region linked to associative learning. The model demonstrates robust performance in pattern recognition, implying a similar encoding mechanism in insects. Integrating behavioural, neurobiological, and computational insights, this study highlights how spatiotemporal coding in the lobula efficiently compresses visual features, offering broader insights into active vision strategies and bio-inspired automation.

*For correspondence: maboudi@gmail.com

## Editor's evaluation

Inspired by bee's visual behavior, this manuscript develops a model of visual scanning, processing, and pattern recognition learning. The work shows how pre-training with natural images creates spatiotemporal receptive fields in lobula neurons that enhance pattern discrimination through sparse encoding. The authors provide a solid analysis of neural responses, model performance across tasks, and the contributions of components like scanning strategies and lateral inhibition. While the model represents a functional circuit for active vision, its biological plausibility is somewhat limited by intentional simplifications. The systematic evaluation of necessary components and comparisons with bee

behavioral data strengthen the findings. This important work offers insights into motion-driven visual processing in compact neural systems.

## Introduction

Bees are capable of remarkable cognitive feats, particularly in visual learning (*Chittka, 2022*; *Srinivasan, 2010*; *Turner, 1911*; *von Frisch, 1914*; *Wehner, 1967*). They can not only learn to associate a colour or orientation of a bar with reward (*Dyer et al., 2011*; *Guiraud et al., 2025b*; *MaBouDi et al., 2020b*; *Srinivasan, 1994*; *Stach et al., 2004*) but are also able to identify specific features to categorise visual patterns, by finding the relevant stimuli properties (*Benard et al., 2006*; *Guiraud et al., 2025a*; *Stach et al., 2004*). Furthermore, bees have demonstrated the capacity to grasp abstract concepts (*Avarguès-Weber et al., 2011*; *Giurfa et al., 2001*; *Guiraud et al., 2018*; *MaBouDi et al., 2020c*; *Menzel, 2012*) and solve numerosity tasks by sequentially scanning the elements within a stimulus (*MaBouDi et al., 2020a*). These exceptional capabilities position bees as a valuable animal model for investigating the principles of visual learning through the analysis of their behavioural responses (*Menzel and Giurfa, 2006*; *Srinivasan, 2010*). Nevertheless, it remains unclear how bees, despite their supposedly low visual acuity (*Gribakin, 1975*; *Spaethe and Chittka, 2003*; *Srinivasan and Lehrer, 1988*) and limited neural resources, recognise complex patterns and perceive the intricacies of the natural world encountered during foraging (*Chittka and Niven, 2009*; *Giurfa, 2013*).

The natural scene that animals encounter is structured differently from random or artificial ones (*Clark et al., 2014*; *Ruderman and Bialek, 1994*; *Simoncelli and Olshausen, 2001*; *Zimmermann et al., 2018*). It has been hypothesised that visual sensory neurons evolve to exploit statistical regularities in natural scenes, efficiently encoding information through their spatiotemporal structures (*Barlow, 1961*). Over evolutionary time, insect visual neurons have developed mechanisms that provide robust and efficient responses to naturalistic inputs (*Dyakova et al., 2019*; *Dyakova et al., 2015*; *Dyakova and Nordström, 2017*; *Juusola et al., 2025*; *Song and Juusola, 2014*; *Zheng et al., 2006*). For instance, *Song and Juusola, 2014* showed that fly photoreceptors extract more information from naturalistic time series than from artificial stimuli or white noise, yielding stronger responses with a higher signal-to-noise ratio (*Song and Juusola, 2014*). Additionally, numerous studies have demonstrated that insect sensory pathways and their associated behaviours dynamically adapt to varying environmental conditions, adjusting their responses based on input parameters such as contrast, spatial frequency, and spatiotemporal correlations (*Arenz et al., 2017*; *Brinkworth and O'Carroll, 2009*; *Clark et al., 2014*; *Dyakova et al., 2019*; *Dyakova and Nordström, 2017*; *Juusola et al., 2025*; *Juusola and Song, 2017*; *Schwegmann et al., 2014*; *Serbe et al., 2016*; *Song and Juusola, 2014*; *Song and Juusola, 2014*; *van Hateren, 1997*; *van Hateren, 1992*). Experience-dependent adaptation has been observed in fly photoreceptors and motion-sensitive neurons in the lobula plate, enabling efficient visual processing under varying conditions. For instance, photoreceptors adapt their response dynamics to different light intensities, optimising sensitivity to natural stimuli (*Juusola and Hardie, 2001a*; *Juusola and Hardie, 2001b*; *Juusola and de Polavieja, 2003*). Similarly, motion-sensitive neurons such as the H1 neuron adjust their response properties based on prior motion exposure, enhancing motion detection in dynamic environments (*Maddess et al., 1985*). This dynamic plasticity allows insects to process ecologically relevant information in real time. However, the precise neural mechanisms underlying natural scene processing remain elusive and require further investigation. Here, we examine how insect visual circuitry has adapted to regularities in natural scenes, focusing on the efficient coding strategies and robust response mechanisms that enhance visual pattern recognition.

In animal vision, active sampling strategies—wherein animals actively scan their environment to extract visual information over time—are widely observed across species (*Land, 1999*; *Land and Nilsson, 2012*; *Severance and Washburn, 1907*; *Varella et al., 2024*; *Washburn, 1908*; *Washburn, 1916*; *Yarbus, 1967*). Primates employ eye movements, including saccades and microsaccades, to enhance fine spatial resolution and improve the encoding of natural stimuli (*Anderson et al., 2020*; *Land, 1999*; *Näher et al., 2023*; *Rucci et al., 2007*; *Rucci and Victor, 2015*). Similarly, insects utilise active vision strategies, incorporating characteristic head and body movements or specific approach trajectories to optimise visual processing during behavioural tasks (*Bertrand et al., 2021*; *Chittka and Skorupski, 2017*; *Dawkins and Woodington, 2000*; *Egelhaaf et al., 2009*; *Land, 1973*; *Land*

*and Nilsson, 2012*; *Langridge et al., 2021*; *MaBouDi et al., 2025*). Recent studies have shown that *Drosophila* generate photomechanical photoreceptor microsaccades and can move their retinas to stabilise their retinal images, achieving hyperacute vision and enhancing depth perception (*Fenk et al., 2022*; *Hardie and Franze, 2012*; *Juusola et al., 2017*; *Kemppainen et al., 2022b*). Likewise, honeybee vision may require sequential sampling and integration of colour information due to their limited ability to discriminate between similar hues in brief flashes (<50 ms) (*Nityananda et al., 2014*). To overcome this constraint, bees engage in systematic scanning movements, continuously sampling their surroundings to construct an internal neural representation of their environment (*Boeddeker et al., 2015*; *Collett et al., 1993*; *Doussot et al., 2020*; *Guiraud et al., 2018*; *Kemppainen et al., 2022a*; *Langridge et al., 2021*; *Lehrer and Collett, 1994*; *MaBouDi et al., 2020a*; *Werner et al., 2016*). For instance, bumblebees enumerate visual elements sequentially rather than processing them in parallel, suggesting a reliance on scanning behaviour for feature extraction parallel (*MaBouDi et al., 2020a*), and their flight trajectories further indicate that they prioritise specific pattern regions before making a decision, rather than processing the entire pattern globally (*Langridge et al., 2021*; *MaBouDi et al., 2025*). Given the low-resolution nature of compound eyes and the potentially reduced parallel processing capacity in insects compared to vertebrates, it is likely that bees rely on active vision and sequential sampling to construct a more robust neural representation of their environment (*Chittka and Skorupski, 2017*; *Nityananda et al., 2014*). These active strategies, akin to primate eye movements, play a crucial role in early visual processing, redundancy reduction, and efficient encoding of visual stimuli (*Doussot et al., 2020*; *Kuang et al., 2012*; *Odenthal et al., 2020*). However, it remains poorly understood how such mechanisms allow bees to overcome representational constraints, detect visual regularities, and solve complex discrimination tasks. Understanding these strategies is key to uncovering the fundamental principles of insect vision and their broader implications for visual processing across biological and artificial systems.

Building on our previous work analysing bee flight paths during a simple visual task (*MaBouDi et al., 2025*), we further investigated the main circuit elements that contribute to active vision in achromatic pattern recognition, focusing on a simplified yet biologically plausible model. Our primary objective was to determine how bees' scanning behaviour contributes to the functional organisation and connectivity of neurons in the visual lobes. We hypothesised that the bees' scanning behaviours have adapted to sample complex visual features in a way that efficiently encodes them into spatio-temporal patterns of activity in the lobula neurons, facilitating distinct and specific representations that support learning in the bees' compact brain. To test this, we developed a neuromorphic model of the bee optic lobes incorporating efficient coding principles via a novel model of non-associative plasticity. This model demonstrates how spatial scanning behaviour in response to naturalistic visual inputs has shaped the connectivity within the medulla (the second optic ganglion) to facilitate an efficient representation of these inputs in the lobula (the third optic ganglion). This efficiency is achieved through the self-organisation of a specific set of orientation-selective neurons in the lobula, highlighting the combined impact of scanning behaviour and non-associative learning on shaping the neural circuitry within the bees' optic lobes.

To evaluate the proposed visual network, we enhance our visual processing framework by incorporating a secondary decision-making module inspired by insect associative learning mechanisms, grounded in previous neurobiological evidence (*Cassenaer and Laurent, 2012*; *Fiala and Kaun, 2024*; *Fisher et al., 2015*; *Li et al., 2017*; *Okada et al., 2007*; *Paulk et al., 2009*; *Paulk and Gronenberg, 2008a*). Visual input and flight dynamics for the model were derived from our observations of bee behaviour during a visual discrimination task (*MaBouDi et al., 2025*). This allowed us to evaluate and test the hypothesis of active sampling from our model against real-world behaviour results (*MaBouDi et al., 2025*), as well as other published visual discrimination tasks performed by bees (*Benard et al., 2006*; *Dyer et al., 2005*; *Guiraud et al., 2022*; *Srinivasan, 2010*; *Srinivasan, 1994*; *Zhang and Horridge, 1992*). Furthermore, we conducted a detailed analysis comparing the neural response features emerging from our model with existing neurobiological findings (*James and Osorio, 1996*; *Paulk et al., 2008b*; *Seelig and Jayaraman, 2013*; *Maddess and Yang, 1997*). This alignment enhances the credibility of our model in capturing essential aspects of neural processing underlying active vision.

## Results

### A bio-inspired neural network for active vision

To investigate how bee scanning behaviour optimises neural activity in the visual lobes and enhances visual information processing for efficient pattern recognition, we developed a neural network model inspired by the key morphological and functional characteristics of the bee brain (*Figure 1A, C*). The network abstracts the circuitry responsible for the initial processing of visual input in the bee's lamina and medulla (the first and second optic ganglia). To mimic temporal encoding during scans (*Figure 1B*), we introduced a progressive time delay of 1–5 'temporal instances' between the outputs of medulla neurons and wide-field neurons in the lobula (the third optic ganglion; *Figure 1D*, see Methods). This temporal structuring facilitates sequential sampling of specific locations along the scan trajectory, gradually integrating visual information into a coherent internal representation that emerges as the final output of the lobula neurons.

Building on findings from bee scanning behaviour (*MaBouDi et al., 2025*), the model extracts image input in five sequential patches of 75 × 75 pixels, sampled at a speed of 0.1 m/s, corresponding to a lateral displacement of 15 pixels between consecutive patches (*Figure 1B*; see Methods for details). The green pixel intensities of each patch modulate the membrane potentials of 5625 (75 × 75) grid photoreceptors within the simulated bee's single eye. These photoreceptor responses converge onto 625 lamina neurons via recurrent neural connectivity, providing a feedforward mechanism for transferring visual information. The lamina neurons then project to 250 small-field medulla neurons through a simple feedforward pathway (*Figure 1C*, see Methods).

For each of the five sequential patches that compose a full scan, medulla neuron responses are computed using a spiking neural model. These responses are progressively integrated into the synapses of their corresponding lobula neuron with a structured time delay. As depicted in *Figure 1D*, the synaptic weights dynamically encode the visual information at different temporal instances (T, 2T, 3T, 4T, and 5T), effectively aligning sequentially sampled spatial information into a temporally coherent representation. This ensures that the lobula neuron accumulates and processes the underlying medulla input signals at a synchronised time point, mirroring mechanisms that may occur in biological systems (see Discussion). Additionally, lateral inhibitory connections (red connections in *Figure 1C*) are proposed between lobula neurons to reduce correlation between them, enhancing redundancy reduction in the process.

It is important to note that this proposed spatiotemporal coding is a simplification. In the bee brain, similar processes are likely mediated through dendritic and synaptic latencies, as well as intermediate neuron transmission within the medulla, influenced by non-associative learning in the visual lobe (*Figure 1C, D*). We hypothesise that connectivity in the medulla and lobula can be refined through exposure to sequences of time-varying images, incorporating non-associative learning rules and efficient coding principles. These mechanisms are optimised and shaped through a generative learning process to align with the statistical properties of natural scenes, enhancing the system's capacity for processing complex visual inputs (see next section) (*Figures 2 and 3*).

The neural representation of the visual inputs was subsequently transmitted and processed in the mushroom body—the learning centre of the bee brain (*Ehmer and Gronenberg, 2002*; *Li et al., 2017*; *Paulk et al., 2008b*; *Paulk and Gronenberg, 2008a*; *Schmalz et al., 2022*; *Figure 1C*). To simplify the model, we incorporated a single mushroom body output neuron (MBON), whose firing rate reflects the simulated bee's preference for a given visual input. By adjusting synaptic weights within the mushroom body, the network was trained to classify visual patterns as either positive (low MBON firing rates) or negative (high MBON firing rates; see Discussion). Following non-associative learning and extensive exposure to natural images, the entire network was trained and tested on various pattern recognition tasks commonly used in experimental studies (*Benard et al., 2006*; *Dyer et al., 2005*; *Srinivasan, 2010*; *Srinivasan, 1994*; *Zhang and Horridge, 1992*), including the discrimination of 'plus' and 'multiplication sign' patterns, as previously examined in real bumblebees (*MaBouDi et al., 2025*; *Figure 4A*).

To evaluate the performance of the active vision model, we analysed MBON activity, which functions as a decision-making unit. A lower MBON response from its baseline activity to a particular pattern indicates preference, whereas a higher response suggests rejection. After multiple training trials through a novel associative learning (see Methods and Discussion), the MBON exhibited a distinct response pattern, with reduced activity towards the chosen visual stimulus and increased

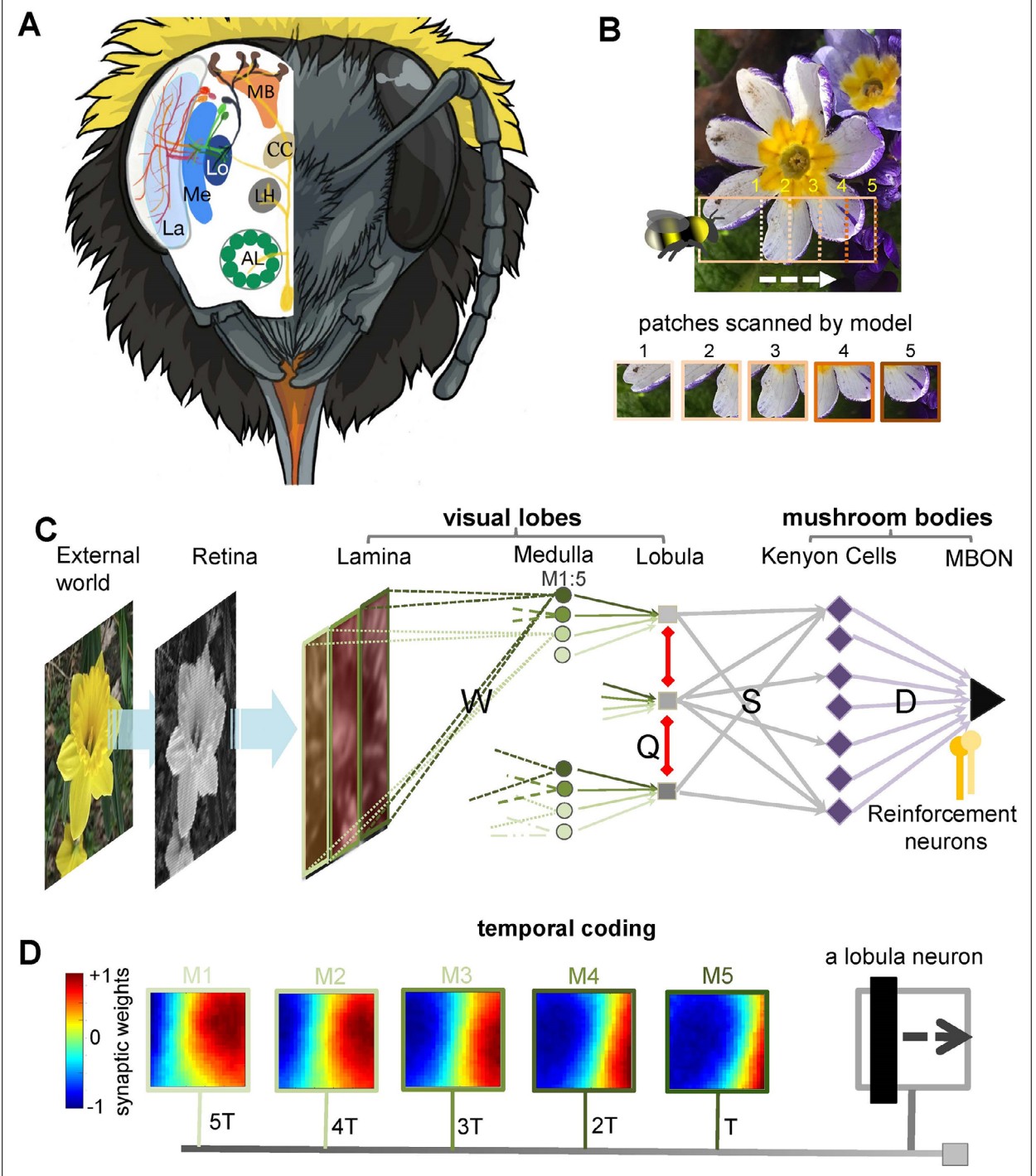

**Figure 1.** Neural network of active vision inspired by neurobiology and flight dynamics of bees. (**A**) The right side displays the front view of the bumblebee head showing the component eye and antenna. Left-hand side presents a schematic view of the bee's brain regions. Part of neural pathways from the retina to the mushroom bodies is also represented. Labels: AL, antennal lobe; LH, lateral horn; CC, central complex; La, lamina; Me, medulla; Lo, lobula; MB, mushroom body. Figure was designed by Alice Bridges. (**B**) A representation of the modelled bee's scanning behaviour of a flower demonstrating how a sequence of patches project to the simulated bee's eye with lateral movement from left to right. Below are five image patches sampled by the simulated bee. (**C**) Representation of the neural network model of active vision inspired by micromorphology of the bee brain that underlie learning, memory, and experience-dependent control of behaviour. The photoreceptors located in the eye are excited by the input pattern. The activities of photoreceptors change the membrane potential of a neuron in the next layer, lamina. The lamina neurons send signals (through $W$ connectivity matrix) to the medulla neurons to generate spikes in this layer. Each wide-field lobula neuron integrates the synaptic output of five small-field medulla neurons. The lobula neurons are laterally inhibited by local lobula interconnections (via $Q$ connectivity matrix). Lobula neurons project their

*Figure 1 continued*

axons into the mushroom body, forming connections with Kenyon cells (KCs) through a randomly weighted connectivity matrix, *S*. The KCs all connect to a single mushroom body output neuron (MBON) through random synaptic connections *D*. A single reinforcement neuron (yellow neuron) modulates the synaptic weights between KCs and MBON by simulating the release of octopamine or dopamine when presented with specific visual stimuli (see Methods). (**D**) A temporal coding model that is proposed as the connectivity between medulla and lobula neurons. Each matrix shows the inhibitory (blue) and excitatory (red) connectivity between lamina neurons to a medulla neuron at a given time delay. In this model, the five small-field medulla neurons that are activated by the locally visual input, at different times of scanning, send their activities to a wide-field lobula neuron with a synaptic delay such that the lobula neuron receives all medulla input signals at the same instance (i.e. in the presented simulation the lobula neuron is maximally activated by the black vertical bar passing across the visual field from the left to right. Each underlying medulla neuron encodes the vertical bar in a different location of the visual field).

activity towards the rejected one. This suggests that the MBON plays a role akin to a decision neuron in pattern recognition tasks. Importantly, no reinforcement learning, or synaptic updates were applied during the testing phase, ensuring that the observed responses reflected the network's learned capacity for visual discrimination rather than online adaptation.

## Non-associative learning shapes spatiotemporal coding in the lobula to align with the statistical features of natural scenes

The synaptic weights in the optic lobe were updated through exposure to natural images during the model's lateral scanning process (see Methods, *Figure 1B*). While lamina-to-medulla connections are structured based on temporal coding (*Figure 1D*), lobula neurons are configured to laterally inhibit each other, facilitating competitive interactions. Synaptic connections were updated using Oja's implementation of Hebb's rule (*Oja, 1982*). Simultaneously, a symmetric inhibitory spike-timing-dependent plasticity (iSTDP) rule was applied to lateral inhibitory connections among lobula neurons (*Vogels et al., 2011*). These local synaptic plasticity rules, which govern interactions between lamina, medulla, and lobula neurons, support non-associative learning—that is synaptic modifications occurring in the absence of reward (see Methods). Together, these plasticity mechanisms drive the network towards an efficient representation of visual input, reducing redundancy while preserving essential visual information.

*Figure 2A* illustrates the receptive fields of lobula neurons, which exhibit spatiotemporal orientation selectivity after training on 100 flower and natural images (comprising 50,000 time-varying image patches). Each square in the figure represents one of the 50 lobula neurons, with the heat map indicating the synaptic weights of the corresponding lamina neurons (connected via the medulla neurons). To aid interpretation, the lower portion of (*Figure 2A*) provides examples of two individual lobula neurons, detailing their lamina synaptic weights for each of the five medulla neurons. For a more dynamic representation of these receptive fields over time, see *Video 1*. The receptive fields of lobula neurons are characterised by an elongated 'on' area (regions representing positive synaptic weights) adjacent to an antagonistic 'off' area (regions with negative synaptic weights). These regions are generally aligned along a specific orientation, and their balance changes dynamically across time-delayed instances of medulla responses. For instance, one lobula neuron responds most strongly to a 135° bar moving orthogonally to its 'on' or 'off' areas while exhibiting little or no response to other orientations (*Figure 2C*). The population of 50 lobula neurons demonstrates specificity to both orientation and direction, closely resembling neuronal responses observed in bees and other insects (*James and Osorio, 1996*; *Paulk et al., 2008b*; *Seelig and Jayaraman, 2013*; *Maddess and Yang, 1997*).

To illustrate how lobula neurons process natural visual inputs, *Figure 2B* depicts the sequence of image patches scanned during a simulated horizontal movement. The results show that only a small subset of lobula neurons respond at any given moment, indicating that their activity is decorrelated and relatively selective—an outcome of non-associative learning mechanisms in the visual lobe (see Discussion). Notably, the two most active lobula neurons captured distinct structural features of the flower petal: one neuron's receptive field aligned with the left 45° edge of the petal, while another matched the right-angled edge. This suggests that the model effectively extracts distinct visual features with a minimal number of filters (lobula neurons).

To examine the selectivity of lobula neurons further, we analysed the spiking activity of a representative neuron tuned to a 150° orientation. As expected, the neuron showed maximal firing (26 spikes/

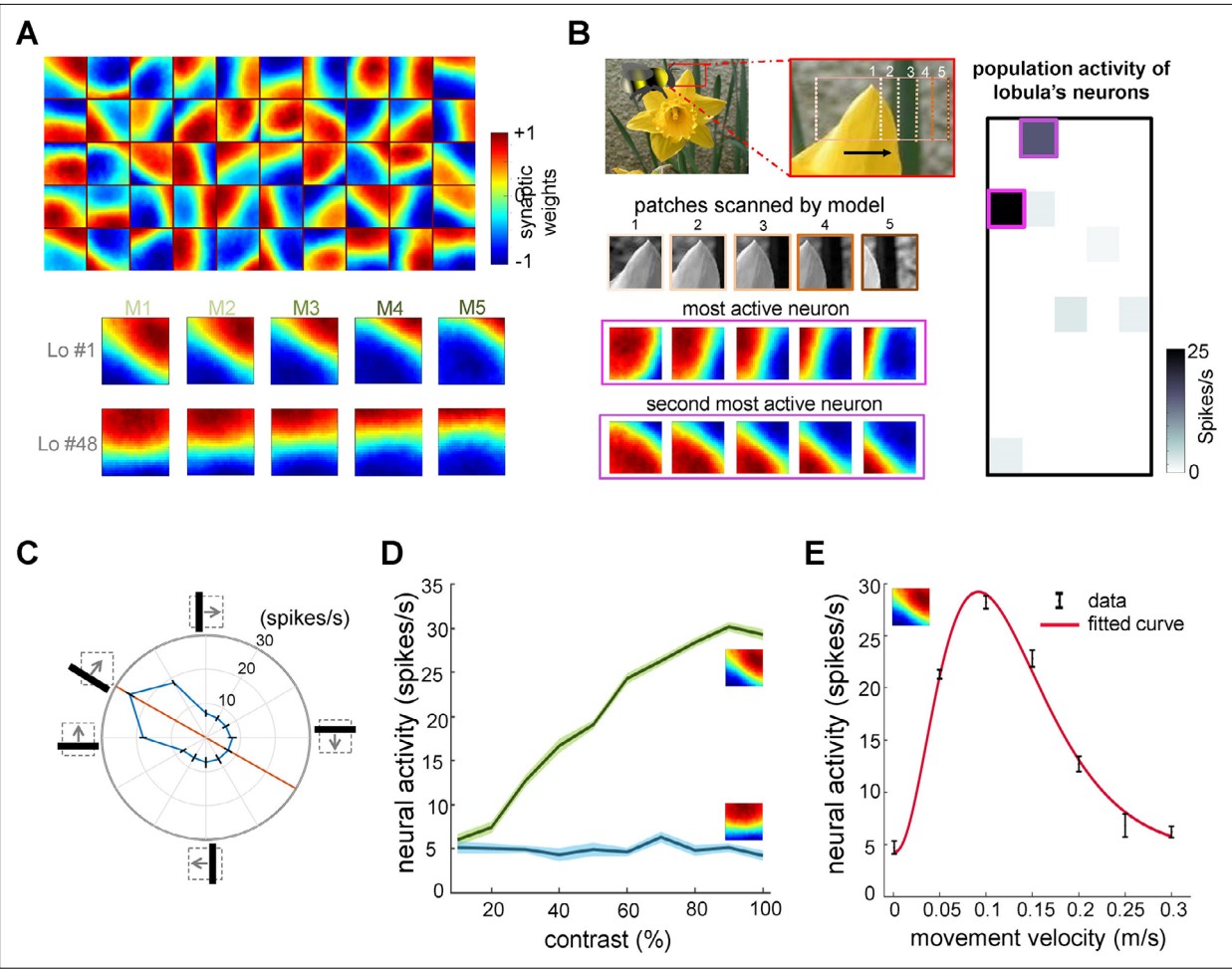

**Figure 2.** Neural responses of the simulated bee model to visual patterns. (**A**) Top: each square in the matrix corresponds to a single time slice of the obtained spatiotemporal receptive field of a lobula neuron (5 × 10 lobula neurons) that emerged from non-associative learning in the visual lobes after exposing the model to images of flowers and nature scenes (see *Video 3*). Bottom: spatiotemporal receptive field of two example lobula neurons are visualised in the five-time delay slices of the matrices of synaptic connectivity between lamina and five medulla neurons (see *Figure 1D*). The lobula neuron integrates signals from these medulla neurons at each of five time periods as the simulated bees scan a pattern (time goes from left to right). Blue and red cells show inhibitory and excitatory synaptic connectivity, respectively. The first example lobula neuron (#1) encodes the 150° angled bar moving from lower left to the upper right of the visual field. The second example lobula neuron (#48) encodes the movement of the horizontal bar moving up in the visual field. (**B**) An example of an image sequence projected to the simulated bee' eye with lateral movement from left to right. Below shows the five images patched sampled by the simulated bee. The right side presents the firing rate of all lobula neurons responding to the image sequence. The spatiotemporal receptive field of two highest active neurons to the image sequence are highlighted in purple. (**C**) The polar plot shows the average orientation selectivity of one example lobula neuron (#1) to differently angled bars moving across the visual field in a direction orthogonal to their axis (average of 50 simulations). This neuron is most sensitive to movement when the bar orientation is at 150°. (**D**) The spiking response of the lobula neuron to the preferred orientation raised as the contrast was increased, whereas the response of the lobula neuron to a non-preferred orientation is maintained irrespective of contrast. (**E**) The average velocity–sensitivity curve (± SEM) of the orientation-sensitive lobula neuron (#1) is obtained from the responses of the lobula neuron to optimal (angle of maximum sensitivity) moving stimuli presented to the model at different velocities. The red line shows the Gamma function fitted to the data.

The online version of this article includes the following figure supplement(s) for figure 2:

**Figure supplement 1.** Emergence of receptive fields in Lobula neurons requires structured natural inputs.

sec) when presented with a 150° moving bar. It also exhibited moderate responses to a horizontal bar and a 120° moving bar (18 spikes/sec) but remained largely unresponsive to other orientations (*Figure 2C*). Consistent with experimental findings (*Maddess and Yang, 1997*), the firing rate of lobula neurons increased with contrast at their preferred orientations, whereas responses to non-preferred orientations remained unchanged across contrast levels (*Figure 2D*).

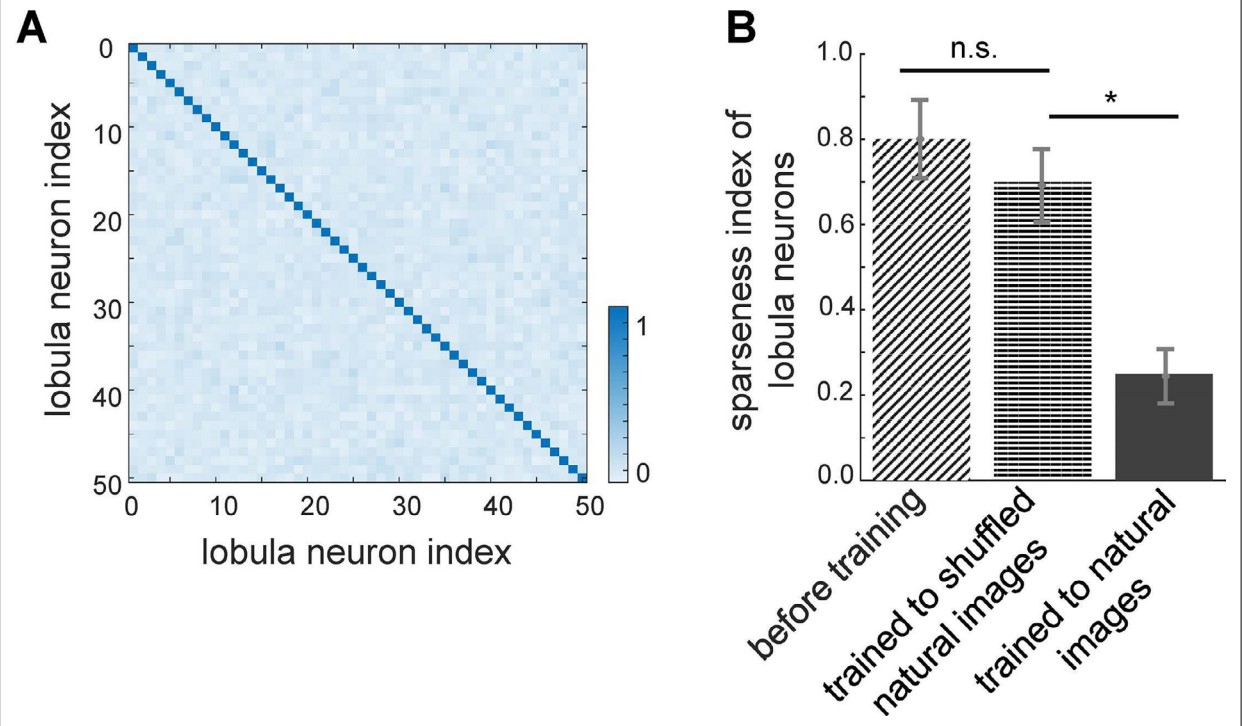

**Figure 3.** Effect of non-associative learning on lobula neuron activity and response sparseness. (**A**) Correlation matrix of lobula neuron responses after training with natural images. The near-diagonal structure indicates that neurons develop distinct and strongly uncorrelated responses, suggesting an efficient, decorrelated representation of visual input. (**B**) Sparseness index of lobula neurons before and after training with different image sets. Before training, neural responses are broadly distributed. Training with shuffled natural images does not change the sparseness of lobula population, whereas training with natural images significantly increases response sparseness, indicating that exposure to structured visual inputs enhances efficient coding. Error bars represent SEM. Asterisks (*) indicate p-values <0.05, while 'n.s.' denotes non-significant results.

Finally, *Figure 2E* highlights the velocity sensitivity of lobula neurons. Each neuron responds maximally at a specific velocity, demonstrating a tuning curve that aligns with known insect neural responses (*Paulk et al., 2008b*; *Maddess and Yang, 1997*). This reinforces that our model successfully captures key quantitative properties of lobula edge detector neurons, including their joint selectivity for orientation, contrast, and motion velocity.

The model demonstrates robustness in generating spatiotemporal receptive fields, even as the number of lobula neurons increases. Training the non-associative learning model with a larger lobula neuronal population while maintaining the same underlying structure from photoreceptors to the medulla, enhances the diversity of orientation-selective responses (see *Figure 2—figure supplement 1A* and *Video 2*). As the number of lobula neurons increases, their tuning properties become more distributed, enabling a finer and more precise encoding of different orientations and motion patterns. This scalability highlights the model's ability to generalise its representation of natural scene statistics while achieving varying levels of resolution in visual encoding.

## Lobula neuron responses become sparse and decorrelated through non-associative learning with natural images

To assess the impact of training on the population activity of lobula neurons, we quantified their response sparsity and decorrelation before and after learning. *Figure 3A* presents the correlation matrix of lobula neurons in response to 10,000 sequential scans of natural images after training. The results reveal a highly decorrelated response pattern, with a strong diagonal structure indicating that each lobula neuron maintains a distinct response profile. This demonstrates that non-associative learning enhances both the selectivity and independence of neural representations, allowing the network to develop more efficient and diverse feature encoding.

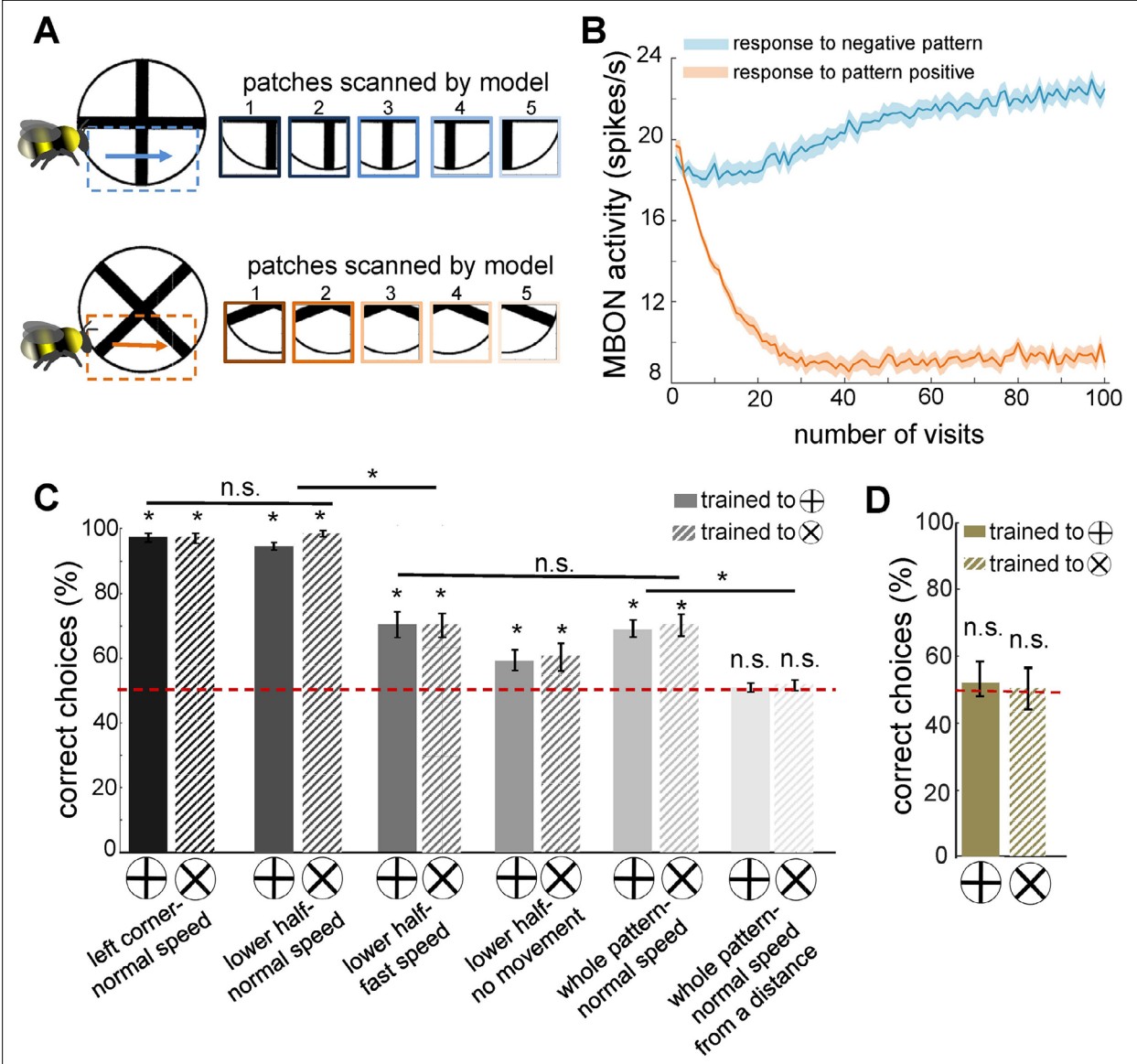

**Figure 4.** Simulated bees' performance in a pattern recognition task using different scanning strategies. Twenty simulated bees, with random initial neuronal connectivity in mushroom bodies (see Methods) and a fixed connectivity in the visual lobe that were shaped from the non-associative learning, were trained to discriminate a *plus* from a *multiplication* symbol (100 random training exposures per pattern). The simulated bees scanned different regions of the patterns at different speeds. (**A**) Top and below panels show the five image patches sampled from the plus and multiplication symbols by simulated bees, respectively. It is assumed that the simulated bees scanned the lower half of the patterns with lateral movement from left to right with normal speed (0.1 m/s). (**B**) The plot shows the average responses of the mushroom body output neuron (MBON) to rewarding multiplication and punishing plus patterns during training procedure (multiplication symbol rewarding, producing an Octopamine release by the reinforcement neuron, and the plus symbol inducing a Dopamine release). This shows how the response of the MBON to the rewarding plus was decreased while its response to the punishing multiplication pattern was increased during the training. The MBON equally responded to both multiplication and plus before the training (at number of visits = 0). (**C**) The performance of the simulated bees in discriminating the right-angled plus and a 45° rotated version of the same cross (i.e. multiplication symbol) (*MaBouDi et al., 2025*; *Srinivasan, 1994*), when the stimulated bees scanned different regions of the pattern (left corner, lower half, whole pattern) at different speeds: no speed 0.0 m/s (i.e. all medulla to lobula temporal slices observed the same visual input), normal speed at 0.1 m/s and fast speed at 0.3 m/s, and from a simulated distance of 2 cm from stimuli (default) and 10 cm (distal view). The optimal model parameters were for the stimulated bees at the default distance when only a local region of the pattern (bottom half or lower left quadrant) was scanned at a normal speed. (**D**) Mean performance (± SEM) of two groups of simulated bees in discriminating the plus from multiplication patterns when their inhibitory connectivity between lobula neurons were not modified by non-associative learning rules. Asterisks (*) indicate p-values < 0.05, while 'n.s.' denotes non-significant results.

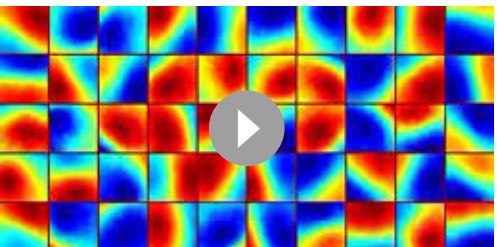

**Video 1.** Spatiotemporal dynamics of receptive fields in 50 lobula neurons emerging from non-associative learning and active scanning. Each square in the matrix represents a single time slice of the spatiotemporal receptive field for a lobula neuron (5 × 10 array of neurons). These receptive fields illustrate the connectivity matrix between five medulla neurons and their corresponding lobula neuron, operating under a temporal coding structure. In this framework, each of the five medulla neurons sequentially transfers a portion of the visual input to the lobula neuron through excitatory (red) and inhibitory (blue) synaptic connections. These receptive fields develop within the visual lobes after the model is exposed to natural images, including flowers and scenery. As the simulated bee scans a visual pattern, lobula neurons dynamically integrate inputs from medulla neurons over time, forming a temporally structured neural representation of the visual scene.

https://elifesciences.org/articles/89929/figures#video1

Further supporting this, *Figure 3B* displays the sparseness index of lobula neurons under different training conditions. Before training, lobula neuron activity was broadly distributed, as reflected in the high sparseness index (see Methods). Training on natural images significantly reduced the sparseness index, indicating that lobula neurons developed a more selective and efficient coding scheme, where only a small subset of neurons responded to any given input. In contrast, training on shuffled natural images led to only a moderate reduction in sparseness, suggesting that the implemented local synaptic plasticity rules alone are insufficient for optimal feature encoding without exposure to structured natural inputs. Notably, while shuffling natural images preserves pixel intensity and overall distribution, it disrupts spatial correlations and higher-order structures present in natural scenes. Consequently, training on these shuffled datasets results in non-structured receptive fields, leading to broadly distributed and less selective coding in the lobula neurons (*Figure 2—figure supplement 1B*).

These findings underscore the role of non-associative learning in shaping neural representations, fostering both sparsity and decorrelation in lobula neurons. Such sparse coding is crucial for efficient sensory processing, as it minimises redundancy while preserving essential visual information. Moreover, it optimises metabolic efficiency in neural networks, aligning with coding strategies observed in biological visual systems (see Discussion).

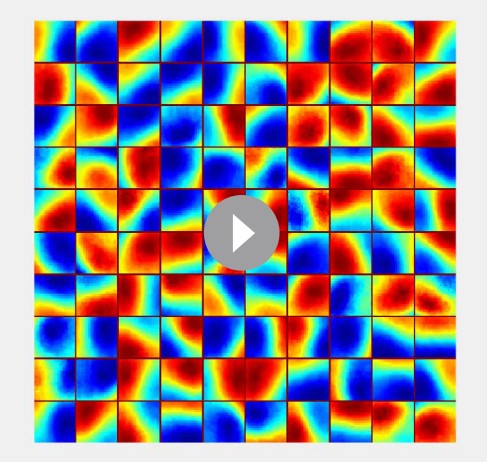

**Video 2.** Spatiotemporal dynamics of receptive fields in 100 lobula neurons emerging from non-associative learning and active scanning. This follows the same structure as *Video 1*, depicting the receptive field evolution in a larger population of 100 lobula neurons under the temporal coding framework and non-associative learning.

https://elifesciences.org/articles/89929/figures#video2

## Active vision enhances visual discrimination through sequential scanning

To replicate bee behavioural findings reported in the literature (*Benard et al., 2006*; *Dyer et al., 2005*; *MaBouDi et al., 2025*; *Srinivasan, 1994*; *Srinivasan, 2010*; *Zhang and Horridge, 1992*), we implemented computational plasticity within the mushroom body circuitry, leveraging sparse lobula neurons that emerge through non-associative learning—critical for encoding both appetitive and aversive values. Specifically, we incorporated classical spike-timing-dependent plasticity (STDP), modulated by dopamine, to regulate synaptic modifications between mushroom body Kenyon cells (KCs) and extrinsic MBONs in response to negative (unrewarded) patterns. Additionally, we introduced a novel STDP-based plasticity rule, modulated by octopamine (see Methods, Figure 9), which we hypothesise induces synaptic depression among KC–MBON connections in response to positive

(rewarded) patterns. These plasticity mechanisms allowed us to explore synaptic dynamics underlying the discrimination of rewarded and non-rewarded stimuli (see Discussion) (*Figure 5*).

The model was trained using a differential conditioning paradigm, where the correct stimulus (S⁺) was paired with a reward and the incorrect stimulus (S⁻) was associated with punishment. For simplicity, we denote positive patterns as S⁺ and negative patterns as S⁻. During associative learning simulations, only the synaptic weights between KCs and MBONs corresponding to the presented patterns were updated, while randomly weighted connections between the lobula and KCs were incorporated to ensure sparse activity in KCs (*Figure 1C*; see Methods). To capture individual variability observed in bees, we repeated the simulations with different initial conditions, including random neural connectivity between lobula neurons and KCs, as well as between KCs and MBONs. The model's performance was then assessed across multiple visual discrimination paradigms (*Figures 4, 6, and 7*).

In the initial implementation, simulated bees were trained to distinguish a plus sign from a multiplication sign, with training focused on the lower half of the plus pattern (*Figure 4A*). Following training, MBON activity decreased in response to the plus (S⁺) while increasing in response to the multiplication sign (S⁻), whereas prior to training, MBON responses to both patterns were similar (*Figure 4B*). This demonstrates that the model successfully discriminated between S⁺ and S⁻ through temporal coding and sequential scanning of the visual pattern (*Figure 4C*). In contrast, a model with fixed, random connectivity in the visual lobe failed to differentiate between the plus and multiplication patterns (*Figure 4D*). This underscores the importance of structured connectivity that emerges in the bee visual lobes through non-associative learning—specifically, the development of spatiotemporal receptive fields in lobula neurons—for successful visual learning.

Our model further revealed that rewarding patterns elicited a reduction in extrinsic neuron responses, while punished patterns led to increased responses (*Figure 4B*), a phenomenon consistent with neural activity recorded from alpha lobe PE1 neurons in the mushroom body (*Okada et al., 2007*). Initially, the simulated bees performed worse than real bees in the plus versus multiplication discrimination task (*Figure 4C*, last pair of bars). Using experimentally derived parameters from bumblebee studies—including an average scanning speed of 0.1 m/s (referred to as normal speed) for whole-pattern scanning and a viewing distance of 20 mm—simulated bees achieved a correct choice rate of only 63% for the plus stimulus and 60% for the reciprocal cross-protocol (averaged over 20 simulations; see *Figure 4C*, fifth pair of bars).

However, when the experimental conditions were adjusted so that simulated bees scanned only the lower half of the patterns or focused on the lower left corner—consistent with real bee behaviour (*MaBouDi et al., 2021b*)—correct choice performance improved significantly, reaching ≥96% and ≥98%, respectively (*Figure 4C*, first two pairs of bars). Conversely, increasing scanning speed (resulting in larger separations between sampled image patches) reduced accuracy to 70%, while stationary simulated bees—those that did not actively scan—achieved only 60% correct choices.

In additional experiments where the model bees were trained from a greater distance (>100 mm), they failed to discriminate between patterns. This aligns with behavioural findings in bees, where they initially select a pattern at random and move closer to scan it before making a decision (*MaBouDi et al., 2025*). Studies show that bumblebees approach stimuli to scan patterns at close range, and their initial approach choices are random (*Guiraud et al., 2018*; *MaBouDi et al., 2021b*). These findings underscore the critical role of active vision in visual learning and discrimination, demonstrating that the model effectively captures key aspects of biological visual processing, including sequential sampling, spatial integration, and plasticity-driven adaptation.

## Scanning strategy modulates lobula neural representations: enhanced selectivity with localised sampling

To further investigate how scanning behaviour influences the distinctiveness of neural representations in the lobula and impacts performance in visual learning tasks, we analysed activity patterns and response magnitudes under two different scanning conditions from the first experiment (*Figure 4C*). Heatmaps of lobula neuron activity (*Figure 5A, B*) illustrate neural responses to two stimulus conditions: scanning only the lower half of the pattern at normal speed (Panel A) and scanning the entire pattern at normal speed (Panel B).

The activity maps indicate that different subsets of lobula neurons responded preferentially to the lower half of the stimulus, exhibiting stronger activation in a smaller neuronal subset. In contrast,

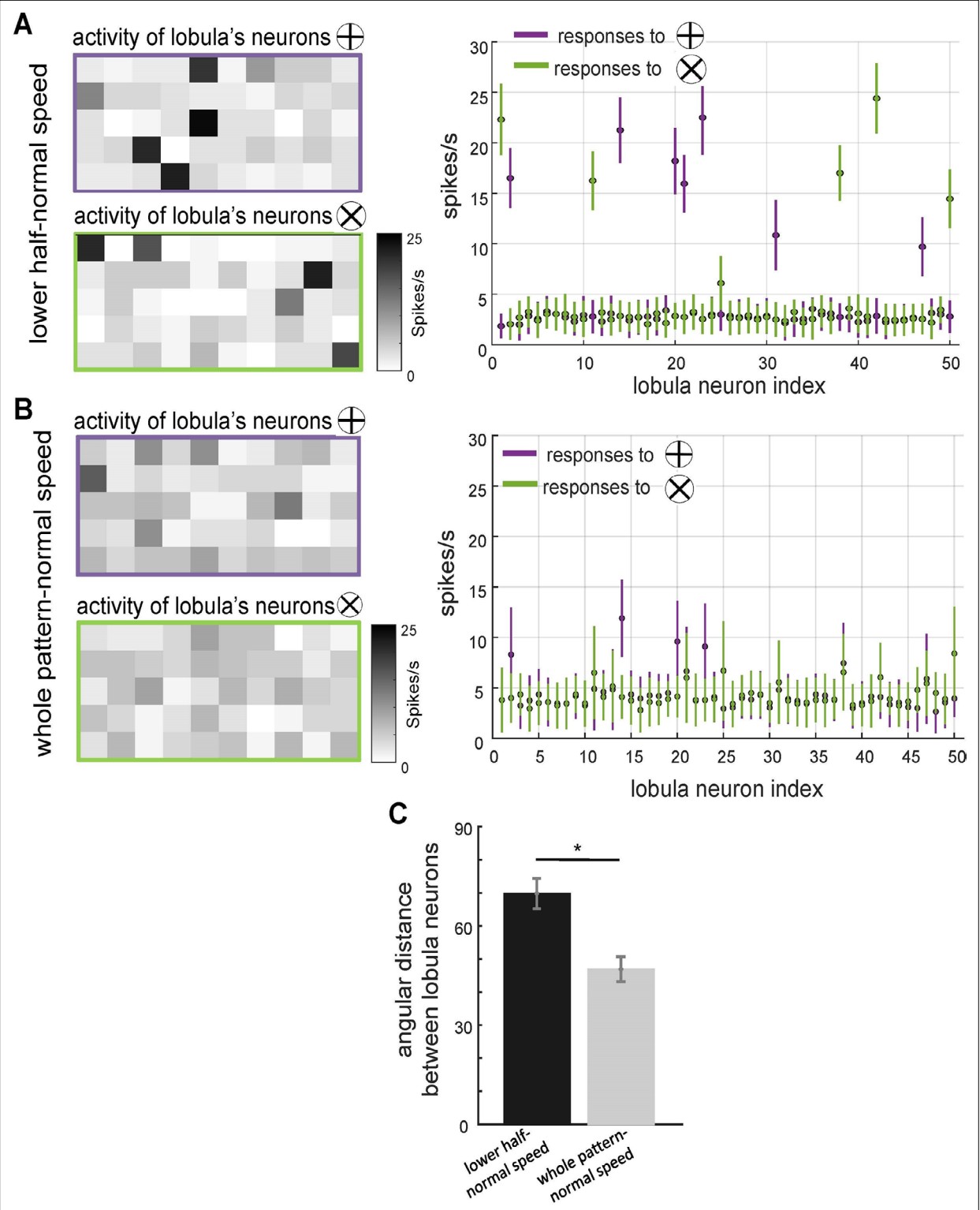

**Figure 5.** The effect of scanning behaviour on the spatiotemporal encoding of visual patterns in lobula neurons responses. (**A, B**) Neural responses of simulated lobula neurons to different scanning conditions. Left panels: heatmaps showing the spiking activity of 50 lobula neurons in response to visual patterns when scanning either the lower half of the pattern at normal speed (**A**) or the whole pattern at normal speed (**B**). Right panels: the mean and standard deviation of the spike rate responses of individual lobula neurons to two distinct visual patterns (plus and multiplication), with colour-coded responses (purple for ⊕, green for ⊗). Scanning behaviour significantly alters the neural responses, with distinct sets of neurons preferentially responding to each stimulus. (**C**) Mean angular distance between lobula neuron responses for different scanning conditions. Lower half-normal speed

*Figure 5 continued on next page*

*Figure 5 continued*

scanning results in greater separation between neural representations, suggesting that scanning of local region enhances feature selectivity. Error bars represent SEM. Asterisks (*) indicate p-values <0.05, while 'n.s.' denotes non-significant results.

scanning the entire pattern resulted in a more widespread and overlapping activation across the lobula neuron population. Corresponding spike rate plots (*Figure 5*, right panels) further highlight these differences, revealing that neurons displayed stronger selectivity when scanning was restricted to the lower half at normal speed compared to whole-pattern scanning. These findings suggest that lobula neurons exhibit stimulus-specific selectivity that is modulated by the spatial dynamics of scanning behaviour.

To quantify the separability of these population responses, we computed the angular distance ($\theta$) between their activity vectors over 20 stimulations using the cosine similarity metric (see Methods). The results (*Figure 5C*) indicate that neural responses exhibited a significantly larger angular distance when the model scanned only the lower half of the stimuli at normal speed compared to scanning the entirety of the same pattern. This suggests that neural activity population were more distinct when scanning was confined to a localised region of these visual patterns, reinforcing the idea that restricted sampling enhances neural discriminability.

These findings demonstrate that lobula neurons encode visual stimuli in a structured manner, with response contrast influenced by both spatial and temporal properties of scanning behaviour (*Figures 4 and 5*). By selectively sampling specific regions of a stimulus, the system enhances the differentiation of these visual patterns, supporting the hypothesis that active vision plays a crucial role in effective neural coding and discrimination.

## Neural network model of active vision bee behaviours across various visual experiments

In this study, we evaluated our model—using scans from the lower half of the visual field—by comparing its performance with results from bee experiments reported in the literature. (Note: Bees may exhibit variations in scanning behaviour under different patterns and training conditions; see Discussion). Our simulated bees successfully discriminated between angled bars (*van Hateren et al., 1990*), a 22.5° angled cross from a 90° rotated version (*Srinivasan, 1994*), and spiral patterns (*Zhang and Horridge, 1992*; *Figure 6A*). When trained on grating patterns with −45° versus +45° orientations, the simulated bees successfully identified the correct pattern. Moreover, it demonstrated the ability to transfer the learned rule to novel patterns the model had never encountered during training, including single-bar patterns (*Figure 6B*). This suggests that the model captures key aspects of visual generalisation observed in real bees. *Figure 6C* shows that the proposed model not only learned to identify the correctly oriented bar pattern but also to distinguish the rewarding pattern from a novel one (two circles). Notably, the model exhibited a 22% lower preference for the negatively trained pattern compared to the novel pattern, validating the implementation of the rejection behaviour and demonstrating that the model can simultaneously learn rewarding and aversive stimuli.

This was further explored by training the network with patterns containing two oriented bars in each lower quadrant (*Figure 6D*; *Benard et al., 2006*; *Stach et al., 2004*; *Zhang and Horridge, 1992*). The simulated bees discriminated these training patterns with over 99% accuracy, but performance dropped to an average of 61% when presented with a simplified variant. When tested with the original positive pattern and novel patterns containing only one correct orientation, the bees showed a high preference for the correct stimulus. Similarly, the simulated bees exhibited a clear preference for a pattern with a single correct feature over the trained negative pattern, indicating that the model can extract multiple features during scanning.

To present a more complex pattern recognition task, we replicated a facial recognition experiment performed on honeybees (*Dyer et al., 2005*) by training the neural network with images of two human faces (*Figure 6E*). As with honeybees, our simulated bees were able to identify the positive trained face from the negative one, as well as distinguish two novel faces and a caricature. Both the real and simulated bees failed to discriminate the faces when rotated through 180°. These results demonstrate that complex visual features can be condensed through spatiotemporal encoding in the lobula neurons into specific and distinct neuronal representations that are critical for learning in the miniature bee brain.

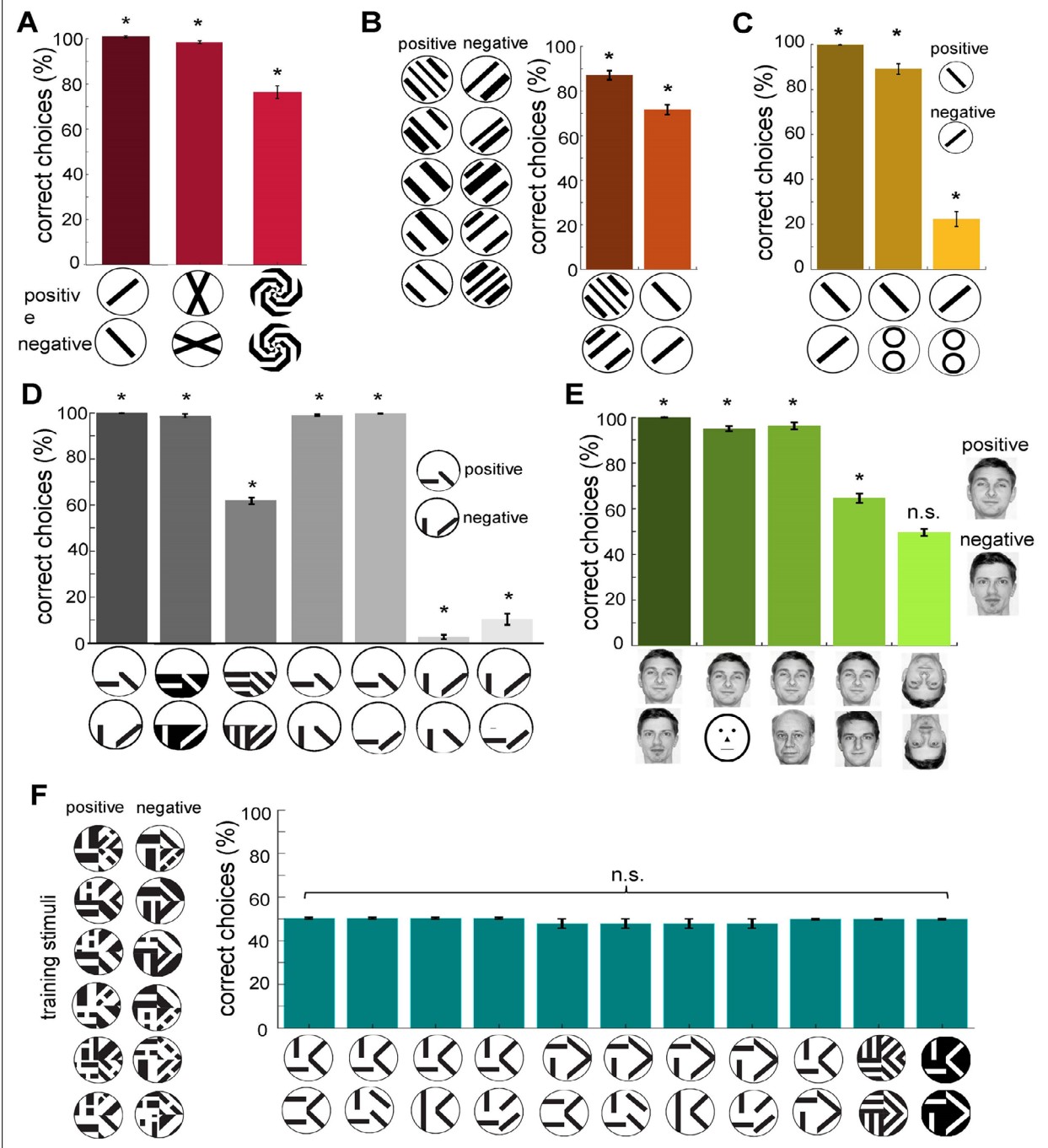

**Figure 6.** Proposed neural network performance to published bee pattern experiments. Twenty simulated bees, with random initial neuronal connectivity in mushroom bodies (see Methods), were trained to discriminate a positive target pattern from a negative distractor pattern (50 training exposures per pattern). The simulated bees' performances were examined via unrewarded tests, where synaptic weights were not updated (average of 20 simulated pattern pair tests per bee). All simulations were conducted under the assumption that model bees viewed the targets from a distance of 2 cm while flying at a normal speed of 0.1 m/s. During this process, the bees scanned the lower half of the pattern. (**A**) Mean percentage of correct choices (± SEM) in discriminating bars oriented at 90° to each other, 25.5° angled cross with a 45° rotated version of the same cross, and a pair of mirrored spiral patterns (*MaBouDi et al., 2025*; *Srinivasan, 1994*). The simulated bees achieved greater than chance performances. (**B**) Performance of simulated bees trained with a generalisation protocol (*Benard et al., 2006*). Trained to 6 pairs of perpendicular oriented gratings (10 exposures per grating). Simulated bees then tested with a novel gating pair, and a single oriented bar pair. The simulated bees performed well in distinguishing between the novel pair of gratings; less well, but still significantly above chance, to the single bars. This indicates that the model can generalise the orientation of the training patterns to distinguish the novel patterns. (**C**) Mean performance (± SEM) of the simulated bees in discriminating the positive orientation from negative orientation. Additionally, the performance in recognising the positive orientation from the novel pattern, and preference for

*Figure 6 continued on next page*

*Figure 6 continued*

the negative pattern from a novel pattern. Simulated bees learnt to prefer positive patterns, but also reject negative patterns, in this case preferring novel stimuli. (**D**) Performance of simulated bees trained to a horizontal and –45° bar in the lower pattern half versus a vertical and +45° bar (*Stach et al., 2004*). The simulated bees could easily discriminate between the trained bars, and a colour inverted version of the patterns. They performed less well when the bars were replaced with similarly oriented gratings, but still significantly above chance. When tested on the positive pattern vs. a novel pattern with one correctly and one incorrectly oriented bar, the simulated bees chose the positive patterns (fourth and fifth bars), whereas with the negative pattern versus this same novel pattern the simulated bees rejected the negative pattern in preference for the novel pattern with single positive oriented bar (two last bars). (**E**) The graph shows the mean percentage of correct choices for the 20 simulated bees during a facial recognition task (*Dyer et al., 2005*). Simulated bees were trained to the positive (rewarded) face image versus a negative (non-rewarded) distractor face. The model bee is able to recognise the target face from distractors after training, and also to recognise the positive face from novel faces even if the novel face is similar to the target face (fourth bar). However, it failed to discriminate between the positive and negative faces rotated by 180°. (**F**) The model was trained on spatially structured patterns from *Stach et al., 2004*, requiring recognition of orientation arrangements across four quadrants. Unlike bees, the model failed to discriminate these patterns, highlighting its limitations in integrating local features into a coherent global representation. Asterisks (*) indicate p-values < 0.05, while 'n.s.' denotes non-significant results.

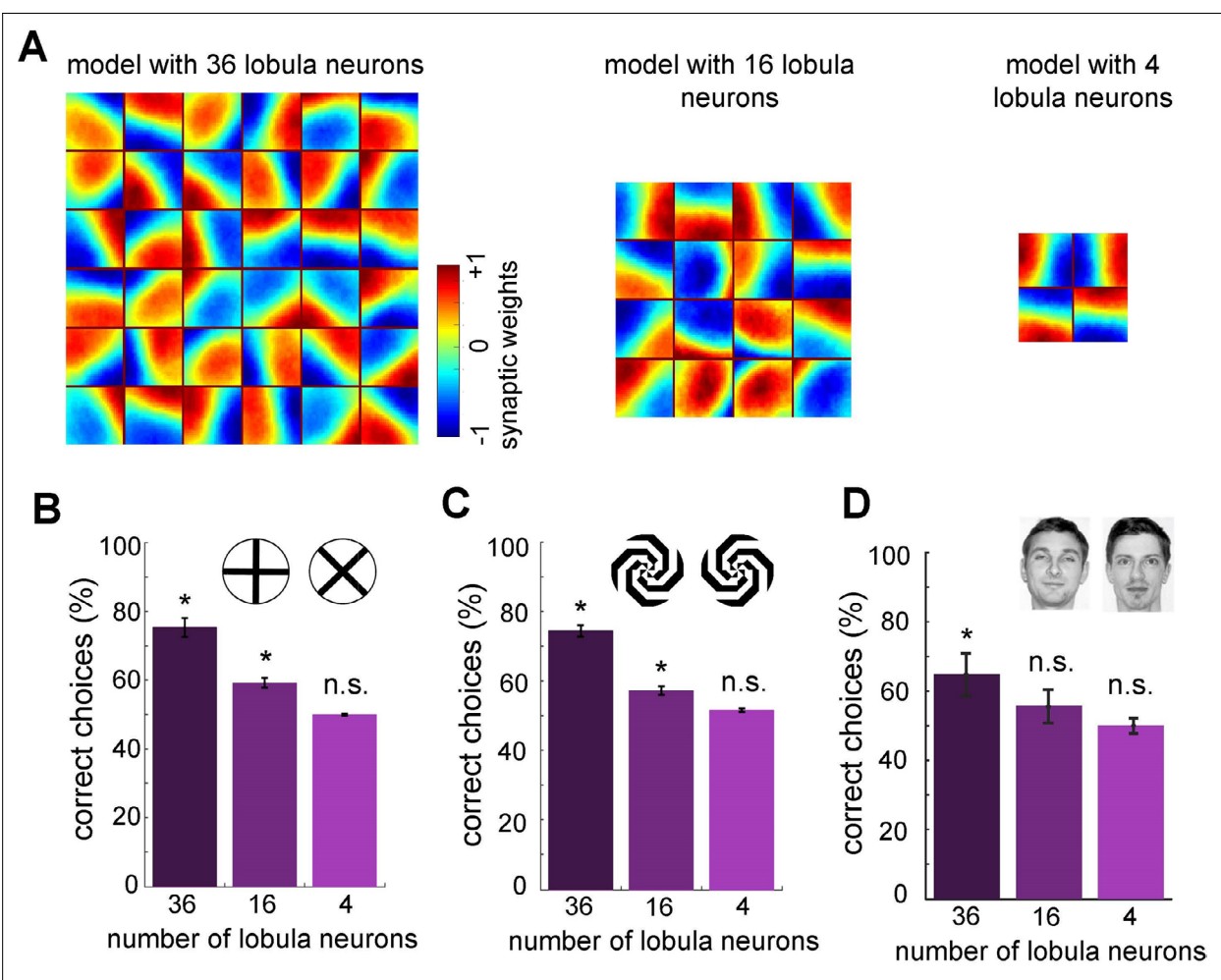

**Figure 7.** Minimum number of lobula neurons that are necessary for pattern recognition. (**A**) Obtained spatiotemporal receptive field of lobula neurons when the number of lobula neurons were set at 36, 16, or 4 during the non-associative learning in the visual lobe (see *Figure 5A*). This shows the models with lower number of lobula neurons encode less variability of orientations and temporal coding of the visual inputs (see *Videos 5 and 6*) The average correct choices of the three models with 36, 16, or 4 lobula neurons after training to a pair of plus and multiplication patterns (**B**), mirrored spiral patterns (**C**), and human face discrimination (**D**). The model with 36 lobula neurons still can solve pattern recognition tasks at a level above chance. It indicates that only 36 lobula neurons that provide all inputs to mushroom bodies are sufficient for the simulated bees to be able to discriminate between patterns. Asterisks (*) indicate p-values <0.05, while 'n.s.' denotes non-significant results.

To evaluate the model's ability to discriminate more complex visual stimuli, we trained it on a set of patterns from *Stach et al., 2004*, which require recognising the spatial arrangement of orientations across four quadrants (*Figure 6F*). Unlike previous experiments where scanning behaviour was confined to specific regions, the complexity of these stimuli necessitated training the model using whole-pattern exposure. The results indicate that the model failed to replicate the bees' ability to discriminate these patterns (*Figure 6F*), despite successfully distinguishing between the plus and multiplication signs when scanning the entire pattern (*Figure 4C*), even if its performance remained lower than that of real bees. Moreover, the model was unable to generalise this capability to more configurations, suggesting that while it can process simple spatial features through sequential scanning, it lacks the longer and more dynamic scanning required for assembling and integrating local features into a coherent global representation, as observed in bees (see Discussion).

## What is the minimally sufficient number of lobula neurons and the necessary connectivity for active vision in bees?

As reported above, our model successfully accomplishes various pattern recognition tasks (*Figures 4 and 6*). We then asked whether our neural networks could perform with a very limited number of lobula neurons that transfer visual information to the mushroom body. To investigate this, we ran the non-associative learning process with different numbers of lobula neurons—specifically, 4, 16, or 36 neurons (the original model had 50). The visual network was subsequently trained using the same set of natural images and protocol as the original model (*Figure 7A*).

Interestingly, the non-associative learning process led to the emergence of distinct spatiotemporal structures in the lobula neurons. We found that reducing the number of lobula neurons decreased the variability in their spatiotemporal receptive (*Figure 7A* and *Videos 3–5*). In particular, when the network was limited to four neurons, it could not encode the full spatiotemporal structure of the training patterns obtained with the model with 50 lobula neurons—only vertical and horizontal receptive fields were produced (*Videos 1 and 5*). As expected, the overall performance of the model decreased as the number of lobula neurons was reduced. Although the model with 16 lobula neurons demonstrated the ability to discriminate more complex patterns beyond the plus and multiplication signs (*Figure 7B, C*), it remained insufficient for recognising highly complex stimuli such as human faces (*Figure 7D, E*).

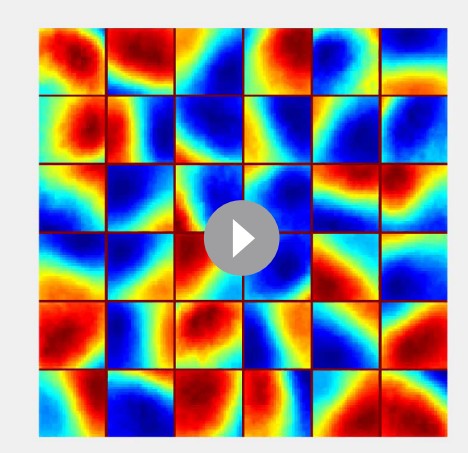

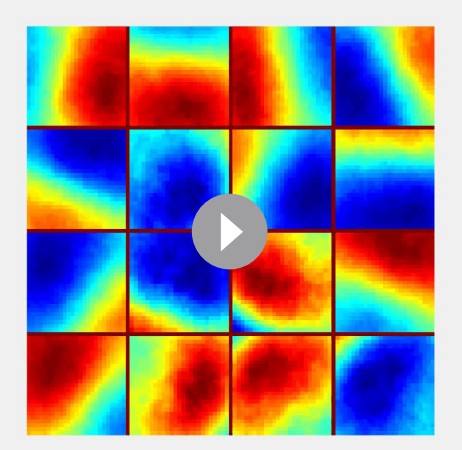

**Video 3.** Spatiotemporal dynamics of receptive fields in 36 lobula neurons emerging from non-associative learning and active scanning. This follows the same structure as *Video 1*, depicting the receptive field evolution in a larger population of 36 lobula neurons under the temporal coding framework and non-associative learning.

https://elifesciences.org/articles/89929/figures#video3

**Video 4.** Spatiotemporal dynamics of receptive fields in 16 lobula neurons emerging from non-associative learning and active scanning. This follows the same structure as *Video 1*, depicting the receptive field evolution in a larger population of 16 lobula neurons under the temporal coding framework and non-associative learning (compare to *Videos 1 and 3*).

https://elifesciences.org/articles/89929/figures#video4

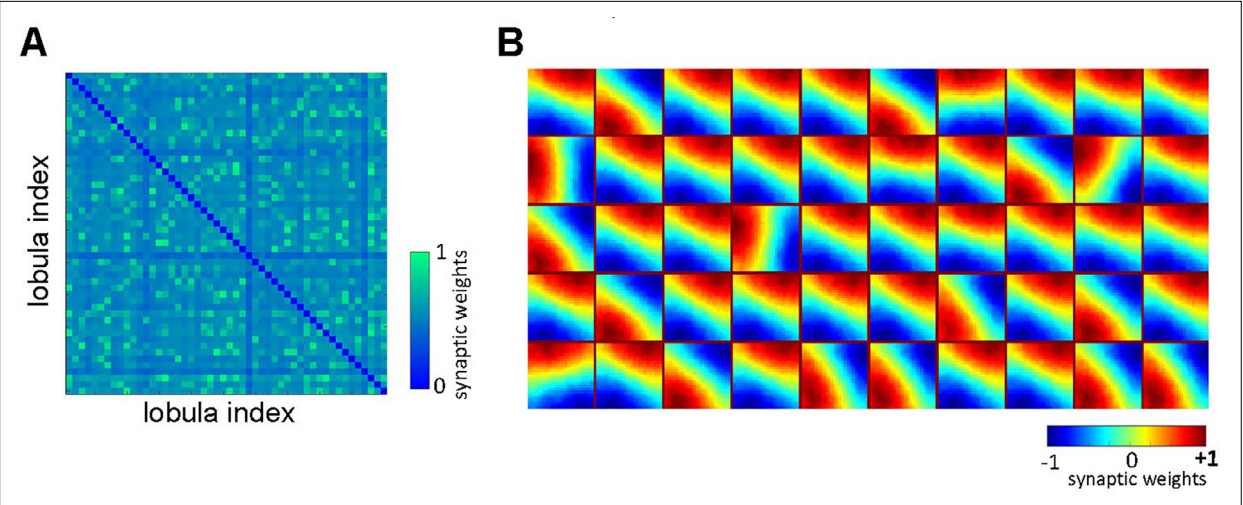

**Figure 8.** The role of lateral inhibitory connections between lobula neurons. Obtained spatiotemporal receptive field of lobula neurons (**B**) when the lateral inhibitory connectivity between lobula neurons is fixed (**A**) during the non-associative learning (see *Figure 2A*, *Videos 1 and 6*).

Moreover, to investigate the effect of inhibitory neurons within the visual lobe on lobula neuron output, we trained the model using the same protocol but fixed the synaptic weights of the inhibitory connections (i.e. these weights were not updated during exposure to the training images). These fixed inhibitory connections limited the ability of the lobula neuron population to encode moving orientations (*Figure 8* and *Video 6*), indicating that the plasticity of inhibitory interneurons in the visual lobe plays a crucial role in facilitating an efficient representation of the visual environment. While this suggests that increasing the network size can enhance the model's capacity for discriminating complex visual patterns, the results (*Figure 4*) indicate that scanning behaviour plays a crucial role in overcoming this limitation. Specifically, adopting a more targeted scanning strategy can improve discriminability by directing visual sampling to the most informative regions of the stimulus (see Discussion).

Taken together, these findings demonstrate that our assumption regarding non-associative plasticity in the visual lobe successfully replicates the neural responses of lobula neurons across various patterns and conditions. This plasticity

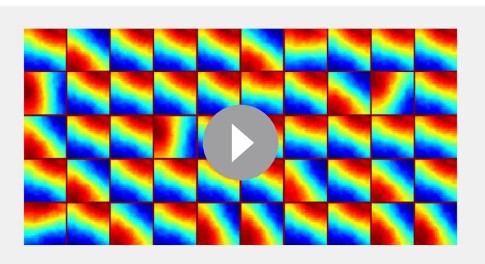

**Video 5.** Spatiotemporal dynamics of receptive fields in only four lobula neurons emerging from non-associative learning and active scanning. This follows the same structure as *Video 1*, depicting the receptive field evolution in a larger population of four lobula neurons under the temporal coding framework and non-associative learning (compare to *Videos 1, 3, and 4*).

https://elifesciences.org/articles/89929/figures#video5

**Video 6.** Spatiotemporal dynamics of receptive fields in 50 lobula neurons emerging from non-associative learning and active scanning. This follows the same structure as *Video 1*, but with fixed lateral inhibitory connectivity between lobula neurons during non-associative learning. The video illustrates how receptive fields evolve under the temporal coding framework, providing a comparison to *Video 1*, where lateral inhibition was plastic.

https://elifesciences.org/articles/89929/figures#video6

yields a sparse, uncorrelated representation of the visual input that benefits subsequent learning processes in the mushroom body. Importantly, these results closely align with theoretical studies (see Discussion), further supporting the effectiveness of the active vision in capturing the underlying principles of information encoding in the insect visual system.

## Discussion

In this study, we investigated the core computational requirements for visual pattern recognition by examined a minimal neural network inspired by the active scanning flight behaviours of bees (*MaBouDi et al., 2025*). We developed a novel model based on the insect visual system, simulating how a small population of lobula neurons encodes the visual environment through spatiotemporal responses. By incorporating non-associative learning, the model self-organises its connectivity within the visual lobe, generating efficient environmental representations (*Figures 2, 3, and 5*). The process leads to the emergence of orientation-selective cells in the lobula, which are essential for encoding complex visual scenes (*Figure 2*).

Our simulations reveal that a small subset of lobula neurons, sensitive to specific orientations and velocities, can condense complex visual environments into spatiotemporal representations expressed as firing rates (*Figure 2*). These sparse representations effectively discriminate between the plus and multiplication patterns used in behavioural experiments (*MaBouDi et al., 2025*) while also generalising to novel stimuli—including successful recognition of human faces—highlighting the model's broader applicability (*Figures 4 and 6*).

Furthermore, our findings highlight the crucial role of bee movement, or active vision, in optimising the analysis and encoding of environmental information (*Figure 4*). Spatiotemporal encoding in the visual lobe emerges as a key mechanism driving the efficiency of minimal intelligent systems. Our study underscores the fundamental computational principles of visual pattern recognition, particularly the interplay between active vision and spatiotemporal encoding in insect information processing. These insights not only advance our understanding of biological vision but also inspire the development of novel computational models for visual recognition tasks (*Figures 3–5*).

The question of how animals cope with a noisy and complex natural world has long been a central topic in neuroscience and behavioural ecology (*Barlow, 1961*; *Gibson, 1978*; *Menzel and Giurfa, 2006*; *Srinivasan, 2010*). One key theoretical framework addressing this challenge is the efficient coding hypothesis, which posits that early sensory systems compress incoming information into a more efficient format, optimising the transmission of relevant signals to higher brain regions (*Barlow, 1961*). According to this hypothesis, individual visual neurons should maximise their output capacity (e.g. reaching their maximum firing rate) when responding to natural stimuli while population responses should exhibit statistical independence (*Simoncelli and Olshausen, 2001*). Despite the relative simplicity of our model compared to recently available full *Drosophila* connectome data (*Lin et al., 2024*; *Schlegel et al., 2024*), our findings suggest that insects optimise visual coding through non-associative learning while actively exploring their environment (see below). Specifically, we demonstrate that neural features of the insect brain, combined with active vision, facilitate this optimisation by developing uncorrelated and sparse coding in the lobula (*Figures 2, 3, and 5*). This supports the idea that efficient coding is not merely a passive process but one actively shaped by an animal's interactions with its surroundings. However, bees exhibit a remarkably diverse behavioural repertoire despite their small brains—ranging from fine-scale object inspections to long-distance navigation (*Chittka, 2022*; *Juusola et al., 2025*; *Menzel, 2012*; *Srinivasan, 2010*). This behavioural diversity makes them an excellent model for investigating how ecological constraints shape neural computation and, ultimately, efficient coding. Understanding how insects dynamically refine sensory representations in response to environmental demands offers broader insights into the fundamental principles of neural information processing in biological systems.

The non-associative model presented in this study operates as a linear generative model that effectively captures the receptive fields of lobula neurons, linking the spatiotemporal statistics of natural environments to principles of efficient neural coding (*Barlow, 1961*; *Olshausen, 2003*). Following training, lobula neuron activity in response to naturalistic spatiotemporal signals becomes highly decorrelated, with only a limited subset of neurons selectively responding to specific visual stimuli (*Figure 5*). This sparse coding strategy enhances energy efficiency by minimising overall neural activity while maintaining distinct stimulus encoding. By ensuring that only a small fraction of neurons is

active at any given moment, this mechanism optimises information processing, reduces redundancy, and enhances metabolic efficiency, reinforcing the adaptive advantages of efficient coding in visual systems. Our model introduces a novel generative framework that can be extended to other species, including primates, to investigate how movement contributes to visual–spatial encoding in larger brains. Given the ubiquity of active vision across the animal kingdom (*Land, 1973*; *Land and Nilsson, 2012*; *Washburn, 1908*; *Washburn, 1916*; *Yarbus, 1967*), the principles identified in this study may be broadly conserved across different taxa, underscoring the fundamental role of sensorimotor interactions in shaping neural representations. These insights provide a valuable foundation for future comparative studies on the interplay between movement, efficient coding, and sensory processing across diverse neural architectures.

Our findings align with previous studies on bumblebees' discrimination of plus and multiplication sign patterns (*MaBouDi et al., 2025*), demonstrating improved model performance when scanning the lower half of patterns at specific velocities (*Figure 3*). However, bees exhibit variations in scanning behaviour depending on pattern complexity and training (*Giurfa et al., 1999*; *Guiraud et al., 2018*). Research has shown that both honeybees and bumblebees solve visual tasks by extracting localised or elemental features within patterns, adapting their discrimination strategies accordingly (*Giurfa et al., 1999*; *MaBouDi et al., 2025*; *Stach et al., 2004*; *Stach and Giurfa, 2005*). This suggests that bees develop tailored flight manoeuvres during training, optimising their scanning behaviour to maximise visual information extraction. Although our model simplifies visual flight dynamics by employing a five-step constant-speed horizontal scan (*Figure 1B*), this abstraction was useful for isolating key computational principles. However, its failure to solve the complex pattern recognition task in *Figure 6F* underscores its limitations, suggesting that incorporating longer and more dynamic scanning strategies could enhance visual processing capacity. Extending the scanning duration while integrating multiple visual features across quadrants—alongside mechanisms such as working memory and sequential learning—could improve performance by enabling the model to retain and integrate previously acquired visual information. Real-world insect vision, however, relies on more flexible and adaptive scanning behaviours shaped by flight speed, head movements, and environmental feedback. Future work should leverage recent advances in insect connectomics, which reveal a diverse range of neuron types—including small object-detecting neurons, motion-sensitive neurons, and colour-processing cells—alongside machine learning techniques for analysing animal movement to develop a more comprehensive flight dynamics model. Incorporating variable scan trajectories and real-time sensorimotor feedback will offer deeper insights into how active vision optimises information acquisition and enhances learning in dynamic environments.

A key advantage of our model lies in its ability to leverage sparsity and selectivity to efficiently process sequential visual data (*Figures 2B, 3 and 6*), in contrast to models that rely on pixel-wise image representation for training. Traditional models that directly process raw pixel values often require substantial computational resources and struggle with scalability, particularly when handling large-scale visual inputs (*Amin et al., 2025*; *Ardin et al., 2016*; *Baddeley et al., 2012*). In contrast, our model extracts and encodes sequential visual information within a small population of lobula neurons (*Figures 2B and 6*), significantly compressing the visual input while preserving essential features. This sparse representation reduces redundancy, enhances computational efficiency, and ensures that only the most informative aspects of the scene are processed for pattern recognition. Furthermore, our model demonstrates adaptive selectivity, dynamically adjusting to different visual inputs by learning lobula responses optimised to the statistical features of the scene. Unlike pixel-based models, which require processing every individual pixel in an image, our approach extracts compact, high-contrast signals that are more robust to noise and enhance generalisability. This is particularly relevant for bio-inspired visual processing, as it aligns with known sparse and decorrelated representations in biological vision systems. By integrating biologically inspired sparse coding, adaptive selectivity, and motion-driven encoding, our model provides a robust alternative to conventional pixel-based architectures. This not only improves computational efficiency but also enhances discriminability and generalisation, making it well suited for real-world applications, including robotic vision and autonomous navigation, where rapid adaptation to dynamic environments is crucial.

The results of our model suggest that passive visual exposure to natural images alters the connectivity in the visual lobes, leading to enhanced pattern recognition abilities (*Figures 4 and 6*). Notably, these synaptic connections develop independently of the initial connectivity profiles of the simulated

bees. We propose that, beyond the general gross neuroanatomy of the insect optic lobes—which has been preserved since the Cambrian period for efficient neural representation (*Ma et al., 2012*)—the specific visual experiences encountered by real bees during early life play a crucial role in shaping their individual visual representations. This, in turn, may influence their subsequent performance in behavioural tasks (*Hertel, 1982*; *Hertel, 1983*; *MaBouDi et al., 2017*; *Vetter and Visscher, 1997*). There is direct empirical evidence for such neural developmental processes in olfactory systems of bees, where early passive exposure improves subsequent odour discrimination (*Arenas and Farina, 2008*; *Locatelli et al., 2013*). Our previous research in olfactory coding demonstrated that the iSTDP learning rule can establish specific connectivity in the sensory system and enhance the separability of odour representations in antennal lobe outputs (*MaBouDi et al., 2017*). A similar mechanism was observed here among lobula neurons, where only a limited subset is activated by specific visual inputs, resulting in sparse and distinct outputs to the mushroom body learning centres (*Figures 2B and 5*). The receptive fields of lobula cells, maintained with the fixed lateral connectivity, shows that inhibition is required for orientation selectivity and temporal coding in the visual lobe (*Fisher et al., 2015*). These findings highlight the critical role of inhibitory connections within the visual lobes. Accordingly, our model predicts that bees with limited early-life visual experiences will perform worse in visual learning and memory compared to bees with rich visual experiences. Further behavioural and neurobiological studies are needed to test this prediction.

Mushroom bodies are critical centres for associative learning and memory in insects (*Fiala and Kaun, 2024*; *Heisenberg, 2003*; *Menzel, 2012*; *Menzel, 2022*). Synapses between KCs and extrinsic mushroom body neurons follow a Hebbian STDP rule (*Aso et al., 2014*; *Cassenaer and Laurent, 2007*; *Markram et al., 1997*); however, the STDP rule alone cannot maintain associative learning (*Abbott and Nelson, 2000*; *Meeks and Holy, 2008*). In insects, associative learning appears to rely on the neurotransmitters octopamine and dopamine to signal unconditioned appetitive and aversive values (*Cognigni et al., 2018*; *Davidson et al., 2023*; *Fisher, 2024*; *Hammer, 1993*; *Hammer and Menzel, 1995*; *Matsumoto et al., 2015*; *Mohammad et al., 2024*; *Perry and Barron, 2013*; *Schwaerzel et al., 2003*; *Selcho, 2024*). These neurotransmitters are released into the mushroom body lobes, where KCs connect to MBON (*Burke et al., 2012*; *Menzel, 2022*; *Okada et al., 2007*; *Strube-Bloss et al., 2011*). Using in vivo electrophysiology in locusts, *Cassenaer and Laurent, 2012*

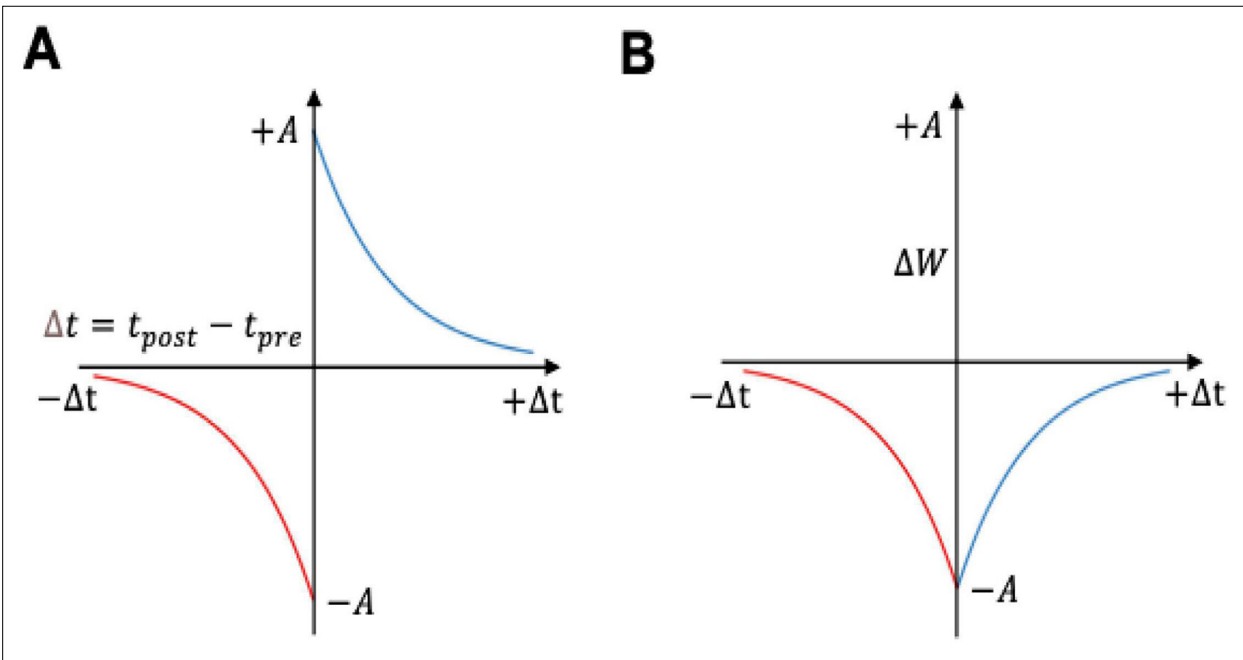

**Figure 9.** Spike-timing-dependent plasticity (STDP) curves. (**A**) Classical spike-timing-dependent plasticity (STDP) curve showing relationship between synaptic weight change and the precise time difference between the Kenyon cells and mushroom body output neuron (MBON) spikes. The synaptic weight can be either depressed or potentiated. (**B**) STDP curve modulated by octopamine in the insect mushroom body. The Synaptic weights are depressed. The formula of these curves is described in *Equations 3 and 4*.

reported that octopamine depresses synapses underlying STDP rule, leading to a lower response in MBONs when octopamine is present.

Following this observation, we modelled associative learning by pairing the positive pattern with the reward via octopamine-modulated STDP (*Equation 4*; *Figure 1*). In this formulation, the temporal ordering of pre- or postsynaptic spikes depresses the synaptic connection between KCs and the MBONS. Conversely, synapses are updated according to classical STDP when negative patterns are paired with the punishment (*Equation 3*; *Figure 9*). This combination produces a complex interplay between synaptic changes and reinforcer signals, enabling the model to not only learn to select the positive patterns but to reject incorrect ones (*Figures 4B and 6C*). The resulting changes in MBON response to positive patterns during associative learning are consistent with the PE1 extrinsic neuron in the honeybee brain, which exhibits a lower response to the positive patterns (*Hancock et al., 2022*; *Fiala and Kaun, 2024*; *Okada et al., 2007*; *Figure 4B*).

However, further studies are required to investigate the novel combination of octopamine and dopamine modulation of STDP that is introduced in this study. Combining non-associative learning in the optic lobes with supervised learning in the mushroom bodies produced a model capable of not only discriminating simple patterns but also generalisation (*Figure 6B*), and correct judgments in conflicting stimulus experiments (*Figure 6D*). The real power of this approach is exemplified in the facial recognition task (*Figure 6E*), where the complexity of the human face is reduced to a set of sparse lobula neuron activations that can be learnt by the mushroom bodies. Moreover, the spatiotemporal receptive fields formed during non-associative learning respond differently for different faces, allowing fine differences to be encoded. Although real bees rarely have to discriminate between human faces, these processes likely enable bees to select rewarding flowers without requiring a complex visual memory within their miniature brains.

We used natural scenes with a statistical structure similar to those that visual systems have adapted to over evolutionary time (*Geisler, 2008*; *Hyvärinen et al., 2009*; *MaBouDi et al., 2016*; *Simoncelli and Olshausen, 2001*). Because bee navigation and foraging primarily involve locating food among a variety of flowers, our non-associative network was trained with a set of different flower images. As with all theoretical models, this is a simplification, since real bees navigate a 3D environment with a large field of view. Here, we assume that receptive field formation in real bees is comparable to our 2D simulations. Nevertheless, further studies are necessary to refine and expand our model based on a more comprehensive understanding of the function and structure of the bee eye components (*Juston et al., 2013*; *Juusola et al., 2017*; *Kemppainen et al., 2022b*; *Viollet and Franceschini, 2010*). Moreover, investigating the neural mechanisms underlying visual learning in the bee brain will allow us to fine-tune our model's architecture and parameters, leading to a more faithful representation of the bee visual system.

In this study, we restricted the model's input to green photoreceptors to align with the known visual processing mechanisms of honeybees. This decision was based on the hypothesis that bee pattern recognition primarily relies on the green component of visual input, as green-sensitive photoreceptors are the most abundant, comprising approximately two-thirds of the ommatidia in the compound eye (*Briscoe and Chittka, 2001*; *Giger and Srinivasan, 1996*; *Spaethe et al., 2001*; *Spaethe and Briscoe, 2004*). There is also empirical evidence that the green channel provides the predominant input to movement and edge detection, as well as detailed spatial information for visual discrimination (*Giger and Srinivasan, 1996*; *Spaethe and Chittka, 2003*). Moreover, natural images exhibit a strong correlation among colour channels, meaning that excluding certain channels does not substantially alter the structure of the visual scene. By focusing on the green photoreceptor input, our model remains biologically plausible while ensuring computational efficiency. Future work could explore the contributions of other photoreceptor types to assess their impact on visual pattern recognition and potential interactions between colour and spatial information processing in the insect visual system.

Our model provides a functional abstraction of the insect visual system, focusing on core computational principles rather than replicating the detailed structural connectivity available from recent connectome studies (*Lin et al., 2024*; *Schlegel et al., 2024*). By prioritising the identification of fundamental mechanisms underlying active vision and visual learning, our approach avoids the challenges associated with highly parameterised models that can be difficult to interpret mechanistically. Integrating known physiological properties and behavioural findings from bees, our model generates testable hypotheses on how motion-driven visual processing enhances pattern recognition. This

functional simplification allows us to isolate key mechanisms that might otherwise be obscured in large-scale anatomical reconstructions. Additionally, our study emphasises the critical role of motion in visual recognition—an aspect often overlooked in static-image studies—and demonstrates how sequential visual input dynamically shapes neural encoding, reinforcing the importance of active vision in efficient sensory processing.

Recent studies have shown that bees often employ efficient, low-cost strategies to solve cognitive tasks (*Cope et al., 2018*; *Guiraud et al., 2018*; *Langridge et al., 2021*; *MaBouDi et al., 2021a*; *MaBouDi et al., 2023*; *MaBouDi et al., 2020b*; *Roper et al., 2017*; *Vasas et al., 2019*). Understanding these cognitive strategies not only advances our knowledge of neural computation in miniature brains but also provides a framework for improving artificial intelligence and autonomous systems. Our study highlights the minimal neural architectures required for visual learning and lays the foundation for bio-inspired, unsupervised machine learning algorithms. By emphasising active vision through movement-driven pattern recognition, our model offers insights into solutions for key AI challenges, such as visual invariance and robust 3D environmental understanding. Moreover, engineering implementations of eye micromovements have been shown to enhance edge and bar discrimination, improving the visual processing efficiency of flying robots (*Juston et al., 2013*; *Viollet and Franceschini, 2010*). Additionally, the non-associative learning model and local plasticity rules explored in this work closely align with unsupervised learning techniques, particularly sparse coding models, where sparsity constraints enhance efficiency by reducing redundancy and promoting selective coding. This bio-inspired framework enables the extraction of latent structures in high-dimensional temporal data, with applications ranging from sensory signal processing to more adaptive and robust autonomous perception. Bridging biological and machine intelligence through evolutionarily optimised computational strategies paves the way for the next generation of AI, driving advancements in robotics, autonomous navigation, and real-world learning systems (*de Croon et al., 2022*; *Manoonpong et al., 2021*; *Serres and Viollet, 2018*; *Webb, 2020*).

## Materials and methods
### Network topology of active vision model

The model architecture of the bee visual pathway is illustrated in *Figure 1A*. The bumblebee has a pair of compound eyes that are composed of ~5500 ommatidia (*Spaethe and Chittka, 2003*; *Streinzer et al., 2013*). Each eye contains three different types of photoreceptors, short, medium and long wavelength sensitive peaking in the UV, blue and the green, respectively (*Menzel and Blakers, 1976*; *Skorupski et al., 2007*). Since the green photoreceptors are those that predominantly mediate visual pattern recognition (*Giger and Srinivasan, 1996*; *Spaethe et al., 2001*), we modelled that $75 \times 75$ green photoreceptors in one eye component are activated by the pixel values of the input pattern. Photoreceptors then project to 625 ($25 \times 25$) neurons in the lamina, which is the first centre of visual processing. In this model, each lamina neuron, $r_l^{La}$, receives input from a non-overlapping $3 \times 3$ grid of neighbouring photoreceptors, corresponding to adjust ommatidia. The response of a lamina neuron is computed as $r_l^{La} = f\left(\sum_{p=1}^{P=9} r_p; A_0, m, b\right)$. Here, the activation function $f$ is defined as:

$$f\left(r; A_0, m, b\right) = A_0 / \left(1 + exp\left(mr + b\right)\right),$$

where $r$ is the input to the lamina neuron $La$, and $A_0$ represent the maximum possible activity of lamina neurons. The parameters $m$ and $b$ define the shape of the activation function, controlling its steepness and midpoint, respectively. For simplicity, the activation function in our model is fixed with $A_0 = 1$, $m = -1$, and $b = 0.5$. This function imposes a constraint where weak inputs result in low activity, while stronger inputs drive the response towards its maximum value in a sigmoidal manner. Each photoreceptor's output, $r_p$, is derived directly from the pixel intensity at the corresponding location in the input image, representing the green channel's brightness. The values are normalised between 0 and 1, ensuring a continuous response that reflects the natural variation in luminance.

In this study, each spiking neuron operates according to the integrate-and-fire model. The dynamics of the subthreshold membrane potential of a neuron, $u(t)$ is described by the following standard conductance-based leaky integrate-and-fire model: $\tau \frac{du(t)}{dt} = -u(t) + R.I(t)$, where $R = 10$

and $\tau = 10\,\text{ms}$ are the resistance and membrane time constant of the neuron, respectively. Here, the input $I(t)$ exhibit the total synaptic input to the cell from presynaptic neurons. The membrane potential is reset to the base activity, $v_0 = -80\,\text{mV}$, if it exceeds the threshold, $V_T = 0\,\text{mV}$.

Each medulla neuron is activated by the summed activity of lamina neurons through the synaptic connectivity matrix W. The input of the m—the medulla neuron, $I_m^{Me}$, is calculated $I_m^{Me} = \sum_{l=1}^{L} W_{m,l} r_l^{La}$. The value $W_{l,m}$ specifies the strength of a synaptic input from the $l$th lamina neuron to the m—the medulla neuron. To account for the inherent variability in neural responses, we incorporated stochasticity by adding signal noise generated from a Poisson distribution to the output of each neuron. The Poisson distribution was chosen because it closely models the statistical fluctuations observed in biological neural firing, where the variance of the response scales with the mean activity. This noise was applied independently to each neuron, ensuring that the variability in responses remains biologically plausible while preserving the overall signal structure. By introducing this element, the model better reflects the natural dynamics of neural processing, capturing the probabilistic nature of spike generation and sensory encoding in biological systems.

We propose a temporal coding model that captures the interaction between medulla and lobula neurons in the visual pathway, incorporating sequential scanning of visual stimuli to optimise information processing. In this model, each wide-field lobula neuron receives synaptic input from $M$ small-field medulla neurons, with a structured progressive delay $T$ (*Figure 1D*). Here, $M = 5$ represents the number of temporal instances within the model's input sequence. Each medulla neuron is activated by visual information sampled from one of $M$ overlapping segments of an image patch, determined by the scanning speed, and follows the hierarchical processing pathway from photoreceptors to lamina neurons (*Figure 1C*). While individual medulla neurons encode only a fraction of the visual stimulus, the lobula neuron integrates input from all medulla neurons to generate spiking activity, thereby forming a holistic representation of the entire visual scene. The synaptic transmission between medulla and lobula neurons incorporates structured temporal delays at distinct instances $(T, 2T, 3T, 4T, 5T)$, ensuring that sequentially acquired visual information is temporally aligned. This results in the synchronised activation of the lobula neuron at a single unified time point, effectively integrating spatially and temporally structured input into a cohesive internal representation. By simulating the dynamic interplay between spatial sampling and temporal integration, this model mirrors the way bees may optimise visual processing through active vision. The resulting alignment of visual signals enhances feature extraction and pattern recognition, providing a biologically plausible mechanism for encoding complex scenes efficiently (*Figures 2, 3 and 6*).

The model incorporates lateral inhibitory connections between lobula neurons (*Figure 1C*, depicted in red), $Q = [q_{i,j}]$, where $q_{i,j}$ represents the lateral connectivity between $i$th and $j$th lobula neurons. This connectivity along with $W_{m,l}$ are updated during a non-associative learning process, to reduce redundancy and decorrelate overlapping inputs (see next subsection). This inhibitory mechanism enhances contrast and improves pattern recognition by selectively amplifying novel spatial features (*Figures 3 and 6*).

The processed visual information is then transformed to the KCs in the mushroom body. The synaptic connectivity matrix $S^{Lo \to KC} = [s_{o,k}]$ determines the excitatory connections between lobula neurons and KCs in the mushroom body, following previous findings on sparse, random connectivity (*Caron et al., 2013*; *Szyszka et al., 2005*). KCs exhibit sparse activity, with fewer than 5% of KCs activated per stimulus, ensuring high selectivity for particular image features (*Honegger et al., 2011*). This sparsity emerges naturally from the random connectivity and thresholder activation dynamics, reinforcing the sparseness in the model. Each connection weight $s_{o,k}$ is randomly initialised from a uniform distribution in the range $[0, w_{max}]$, where $w_{max}$ is a scaling factor ensuring the limitation of input to the KCs. For each simulated bee, the connectivity matrix $S^{Lo \to KC}$ is randomly reinitialised, ensuring that each instance of the model has a unique but statistically comparable connectivity structure. This reinitialisation reflects individual variations in synaptic wiring and allows us to assess the robustness of the model's pattern recognition ability across different randomly generated network configurations.

All KCs project to a single MBON, which is the final output of the model. The input of the of the MBON, $I_{MBON}$, is computed by the KC–MBON connections $D$ such that $I_{MBON} = \sum_{k=1}^{K} D_k r_k^{KC}$, where $r_k^{KC}$ is the spiking activity of the $k$th KCs. Finally, a reinforcement neuron makes reinforcement-modulated

connections with the KCs and MBON in the presence of the positive and negative patterns (see next subsection).

Each neuron type is denoted with a superscript corresponding to its respective processing stage. For example, $r_p$, represents the response of an individual photoreceptor, while $r^{La}$ denotes the activity of a lamina neuron, where 'La' refers to the lamina layer. Similarly, other superscripts—Me, Lo, KC, and MBON—correspond to specific network components: the medulla, lobula, KCs, and MBONs, respectively. This ensures consistency in notation throughout the model description.

## Training the network via a non-associative learning

We trained the model using 50,000 time-varying image patches randomly sampled from a dataset of 100 natural flowers and scene images. During each training step, the model received an input sequence of five sequential 75 × 75-pixel patches, extracted by shifting 15 pixels across the image from the left or right or the reverse orientation (*Figures 1B and 2B*). This patchwise input simulates the sequential scanning behaviour of bees as they explore visual stimuli.

Using the described network architecture, each time-varying patch dynamically drives spiking activity in lobula neurons as the simulated movement progresses. At the start of training, all inhibitory connection strengths $Q$ were initialised with values randomly drawn from a uniform distribution between 0 and 1. The feedforward synaptic weights $W$ were initialised using Gaussian white noise $N(0, 1)$. As training progressed, the evoked neural responses of lobula neurons to time-varying patches were used to iteratively update both the inhibitory weights ($Q$) and feedforward connections ($W$) simultaneously (see Discussion).

After the image presentation, the feedforward weight $W$ is updated according to Oja's implementation of the Hebbian learning rule (*MaBouDi et al., 2017*; *Oja, 1982*) via

$$\Delta W_{i,j} = \gamma r_j^{Me} \left( r_i^{La} - r_j^{Me} W_{i,j} \right) \tag{1}$$

Here, the $r_j^{Me}$ and $r_i^{La}$ represent the activities of the $j$th medulla and $i$th lamina neurons, respectively. The positive constant $\gamma$ defines the learning rate.

At the same time of processing, the lateral inhibitory connectivity in the lobula is modified by iSTDP (*Vogels et al., 2011*). Here, we model *non-associative learning* in the lobula by a symmetric iSTDP between presynaptic of the inhibitory neurons and postsynaptic lobula neurons. In this learning rule, both temporal ordering of pre- or post-synaptic spikes potentiates the connectivity and the synaptic strength of $j$th inhibitory neuron onto $i$th lobula neuron ($Q_{i,j}$) is updated as follows:

$$\Delta Q_{i,j} = \eta \left( r_i^{Lo} * r_j^{In} - \alpha \right) \tag{2}$$

where $r_i^{Lo}$ and $r_j^{In}$ exhibit the mean firing rate of the lobula and inhibitory neurons, respectively. The depression factor $\alpha$ controls the target activity rate of the lobula neurons. Here, $\eta$ is the learning rate. To simplify, a one-to-one connection between the inhibitory and lobula neuron is assumed in the model such that the activity of the $j$th inhibitory neuron is equal to the activity of the $j$th lobula neuron. The training is terminated when the synaptic weights over time are changed less than a small threshold (0.001). In the training process, synaptic weights were constrained within the range $[-1, +1]$ for the $W$ and $[0, +1]$ for $Q$ to ensure stable convergence and to reflect biological limitations in synaptic transmission strength.

## Associative learning in mushroom bodies

To verify if the lobula neurons can reproduce empirical behavioural results in different visual tasks, the model is enriched with associative learning process in the mushroom bodies (a bio-inspired supervised learning). When the training process of the non-associative learning is terminated, we use a reward-based synaptic wright modification rule in KCs–MBON connection (D), such that, if a stimulus is rewarding (i.e. positive), the corresponding synapses between activated neurons will be weakened while for a stimulus paired with punishment (i.e. negative), activated synapses are strengthened (*Cassenaer and Laurent, 2012*) (see Discussion). The model behaves as the activity of mushroom body neurons in decreasing their firing rate in responding to the positive stimuli during training (*Okada et al., 2007*). In this model, two reinforcement neurons modulated strengths of synaptic

connectivity at the output of the KCs in response to both reward and punishment. In the presence of the negative patterns, the synaptic strengths from the KCs to the MBON are modified, and modulated by dopamine, based on the classical STDP (*Song et al., 2000*; *Zhang et al., 1998*; *Figure 9*):

$$STDP_{Dop}\left(\Delta t\right) = \begin{cases} Ae^{-\Delta t/\tau}, \Delta t > 0 \\ -Ae^{\Delta t/\tau}, \Delta t < 0 \end{cases}, \tag{3}$$

where $\Delta t = t_{post} - t_{pre}$ implies the difference between the spike time of pre- and post-synaptic neurons. Further, applying the synaptic plasticity rule modulated by octopamine (octopamine-modulated STDP) observed in the presence of rewarding stimuli to the synapses between KCs and MBON (*Cassenaer and Laurent, 2012*), the change in synaptic weight can be summarised as (*Figure 4B*):

$$STDP_{OCT}\left(\Delta t\right) = \begin{cases} -Ae^{-\Delta t/\tau}, \Delta t > 0 \\ -Ae^{\Delta t/\tau}, \Delta t < 0 \end{cases}, \tag{4}$$

Here, $A = 0.01$ and $\tau = 20\,\text{ms}$ exhibit the maximum magnitude and time constant of the STDP function for the synaptic potentiation or depression.

To train the model in different conditions of scanning, the flight-scan forms of the positive and negative patterns were presented to the model. Each set of flight-scan input contained a set of five patches with size 75 × 75 pixels were selected from the test patterns by shifting 15 pixels over each pattern from the left to right (*Figure 4A*). The numbers of shifted pixels control the speed of scanning. The activity of the MBON was used to assess the performance of the model. Following the training, the performance of the model was calculated from a decrease in firing rate of the MBON to a pattern that had been rewarding and/or an increase in firing rate of MBON to a pattern that had been punishing in training. The bee's final behavioural decision is proposed to come from a simple integration of these different valence-encoding neurons.

## Quantifying neural population sparseness and response separability

To quantify population sparseness in the lobula following training on natural images, we employed the Treves–Rolls sparseness index (*MaBouDi et al., 2017*; *Willmore and Tolhurst, 2001*), defined as:

$$SI = \frac{\left(\sum_{j=1}^{N} r_j/N\right)^2}{\left(\sum_{j=1}^{N} r_j^2\right)/N}$$

where $r_j$ represents the firing rate of the $j$th lobula neuron, and $N$ is the total number of lobula neurons.

This metric provides insight into the population coding strategy of lobula neurons, distinguishing between broad distributed representations and sparse selective responses:

- *Maximum SI = 1 (low sparseness)*: Achieved when all neurons respond equally, indicating a fully distributed code where the entire population is uniformly active across all stimuli.
- *Minimum SI = 1/N (high sparseness)*: Occurs when only a single neuron is active while all others remain silent, reflecting a highly selective encoding scheme.

To further assess the separability between response population of lobula neurons in response to different stimuli, we computed the angular distance ($\theta$) using the cosine similarity formula: $\theta = cos^{-1}\left(R_1.R_2/\left(\left|R_1\right|\left|R_1\right|\right)\right)$, where $R_1 and R_2$ represent the activity vectors of two neural populations, and $\left|R_1\right| and \left|R_2\right|$ denote their responses Euclidean norms. This measure captures the geometric distinction between response patterns, with larger angles indicating greater separability between neural representations of different stimuli. A higher angular distance suggests that the population responses are more distinct, reflecting improved stimulus discrimination within the lobula.

## Simulation and statistical analysis

To assess the model's performance across experiments, we conducted 20 independent simulations for each condition, ensuring statistical reliability and robustness. In each simulation, the synaptic connectivity matrix between lobula neurons and KCs was randomly initialised using a uniform distribution

within a biologically plausible range, mimicking the individual variability observed in real bees. This stochastic initialisation prevented bias in learning outcomes and allowed us to examine how the model generalises across different neural configurations.

Each simulated bee underwent multiple training exposures to visual stimuli. Following training, once the model parameters were fixed at the final stage of training, the model was tested across 50 repetitions per condition with testing patterns to account for variability in responses. The model's performance was evaluated by averaging across simulations, with SEM reported in the figures to provide a statistically robust representation of discrimination accuracy.

Statistical analyses were conducted to compare pattern discrimination across conditions. To maintain clarity and focus on the modelling findings, detailed significance values are not reported in the main text. Instead, figures indicate p-values <0.05 with '*' and non-significant results with 'n.s.'. Since data distributions did not always meet normality assumptions, we used the Wilcoxon signed-rank test for matched data and the Wilcoxon rank-sum test (Mann–Whitney $U$ test) for independent samples. For comparisons across multiple groups, we applied the Kruskal–Wallis test, followed by Dunn's post hoc test when necessary.

## Computing environment

All modelling and visualisation were performed using MATLAB (RRID:SCR_001622) and Python (RRID:SCR_008394). MATLAB was also used for statistical analysis.

## Acknowledgements

We thank Paul Graham and Andrew Barron for valuable comments on manuscript and Alice Bridges for drawing the front view of the bumblebee presented in *Figure 1A*. This research was financed by HFSP programme grant RGP0022/2014, EPSRC program grant Brains-on-Board EP/Poo6094/1 and by Horizon Europe Framework Programme grant NimbleAI. MJ was supported by BBSRC (BB/F012071/1 and BB/X006247/1), EPSRC (EP/X019705/1), and Leverhulme (RPG-2024-016). MGG was supported by ARC Discovery Projects DP230100006 and DP210100740 and Templeton World Charity Foundation Project Grant TWCF-2020-20539.

## Additional information

### Competing interests

Mark Roper: is affiliated with Ben Thorns Ltd. James AR Marshall: is co-founder and Chief Scientific Officer at Opteran Technologies. The other authors declare that no competing interests exist.

### Funding

| Funder | Grant reference number | Author |
|---|---|---|
| Human Frontier Science Program | RGP0022/2014 | HaDi MaBouDi<br>Mark Roper<br>Marie-Geneviève Guiraud<br>Lars Chittka |
| Engineering and Physical Sciences Research Council | EP/Poo6094/1 | HaDi MaBouDi<br>Mikko Juusola<br>Lars Chittka<br>James AR Marshall |
| HORIZON EUROPE Framework Programme | NimbleAI | Lars Chittka |
| Biotechnology and Biological Sciences Research Council | BB/F012071/1 | Mikko Juusola |
| Biotechnology and Biological Sciences Research Council | BB/X006247/1 | Mikko Juusola |

| Funder | Grant reference number | Author |
|---|---|---|
| Engineering and Physical Sciences Research Council | EP/X019705/1 | Mikko Juusola |
| Leverhulme Trust | RPG-2024-016 | Mikko Juusola |
| Australian Research Council | DP230100006 | Marie-Geneviève Guiraud |
| Australian Research Council | DP210100740 | Marie-Geneviève Guiraud |
| Templeton World Charity Foundation | 10.54224/20539 | Marie-Geneviève Guiraud |

The funders had no role in study design, data collection, and interpretation, or the decision to submit the work for publication.

## Author contributions

HaDi MaBouDi, Conceptualization, Resources, Data curation, Software, Formal analysis, Supervision, Validation, Investigation, Visualization, Methodology, Writing – original draft, Project administration, Writing – review and editing; Mark Roper, Formal analysis, Validation, Investigation, Visualization, Methodology, Writing – original draft, Writing – review and editing; Marie-Geneviève Guiraud, Investigation, Methodology, Writing – review and editing; Mikko Juusola, Funding acquisition, Validation, Writing – review and editing; Lars Chittka, Conceptualization, Supervision, Funding acquisition, Project administration, Writing – review and editing; James AR Marshall, Supervision, Funding acquisition, Validation, Project administration, Writing – review and editing

## Author ORCIDs

HaDi MaBouDi  https://orcid.org/0000-0002-7612-6465
Mark Roper  https://orcid.org/0000-0003-1135-6187
Marie-Geneviève Guiraud  https://orcid.org/0000-0001-5843-9188
Mikko Juusola  https://orcid.org/0000-0002-4428-5330
Lars Chittka  https://orcid.org/0000-0001-8153-1732
James AR Marshall  https://orcid.org/0000-0002-1506-167X

## Decision letter and Author response

Decision letter https://doi.org/10.7554/eLife.89929.sa1
Author response https://doi.org/10.7554/eLife.89929.sa2

## Additional files

### Supplementary files
MDAR checklist

### Data availability
All figures were generated during the simulation. The code developed for this research project has been made openly accessible on GitHub (copy archived at *MaBouDi, 2025*).

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
