## [Editor Report]

Inspired by bee's visual behavior, this manuscript develops a model of visual scanning, processing, and pattern recognition learning. The work shows how pre-training with natural images creates spatiotemporal receptive fields in lobula neurons that enhance pattern discrimination through sparse encoding. The authors provide a solid analysis of neural responses, model performance across tasks, and the contributions of components like scanning strategies and lateral inhibition. While the model represents a functional circuit for active vision, its biological plausibility is somewhat limited by intentional simplifications. The systematic evaluation of necessary components and comparisons with bee behavioral data strengthen the findings. This important work offers insights into motion-driven visual processing in compact neural systems.

---

## [Decision Letter]

**Decision letter after peer review:**

Thank you for submitting your article "A neuromorphic model of active vision reveals how spatio-temporal encoding in lobula neurons can aid pattern recognition in bees" for consideration by *eLife*. Your article has been reviewed by 3 peer reviewers, and the evaluation has been overseen by a Reviewing Editor and Claude Desplan as the Senior Editor.

Essential Revisions:

The reviewers agree that the model developed by the authors tackles an important problem in machine learning and visual neuroscience. The model is based on visual scanning, which represents a novel and exciting phenomenology in the bee. Unfortunately, the main conclusions of the work must be watered down until the authors demonstrate that alternative, equally plausible, models of the visual and mushroom body circuits are not sufficient to solve the tasks under consideration. We believe that the manuscript can be a valuable and important contribution to the field if the following weaknesses are thoroughly addressed.

1) The neural circuitry underlying the model does not adequately integrate the wealth of *Drosophila* connectome data that has been published during the past 10 years. While the model is definitely bio-inspired, layers of its architecture are built very differently from the connectivity of real insect brains. As a result, many features of the model's architecture appear to be arbitrary. Clarifications should be provided about circuit-function relationships of the bee MB versus the *Drosophila* MB, and their implications on the model.

2) Given the repeated claims that the authors present the "minimal circuit" required for the visual tasks explored, the work ought to rigorously and systematically assess the necessity and sufficiency of the different components included in the circuitry of the model. In particular, could a simpler learning rule be sufficient to explain discrimination? In what sense is the presented circuit "minimal"? Varying the number of lobula neurons of the model is a good first step, but the same should be done for other components of the model.

3) The presentation of model's results and its interpretation should explain the successes and failures of the model in reproducing the actual behavior. It should include a more in-depth comparison of the performances of the model and real bees.

4) The description of the methodology is incomplete, which prevents a proper interpretation of the model's results.

*Reviewer #1 (Recommendations for the authors):*

The introduction talks about natural scenes such that their specific features are critical to neural processing and pattern recognition. However, the manuscript does not thoroughly assess if the model is particularly suited for handling natural scenes. To demonstrate this, it appears to require using non-natural images for non-associative learning and comparing the performance with the model trained with natural scenes. Otherwise, I would recommend rephrasing the introduction.

Line 217: why does restricting the scan field improve the performance dramatically? This may be intuitively reasonable, but it would be nice to have an explicit explanation based on the model's structure.

Discussion:

The insect lobula is not necessarily only composed of wide-field neurons. It would be nice to have some discussion about how other types of neurons, such as small object-detecting neurons, could contribute to the same visual task.

*Reviewer #2 (Recommendations for the authors):*

Line 1 – title – is it really substantiated in this paper that the spatiotemporal lobula encoding "aids" pattern recognition? Relative to what? Can these tasks not be solved by models such as that in Ardin et al. (2016) that use low resolution pixel values as input to the KC and associate the corresponding sparse code with the MBON for selected images?

Line 6 – abstract – in what sense is the presented circuit 'minimal'? The paper explores reducing the number of lobula neurons, but not any other reduction in complexity.

Line 11 – the alignment to neurobiological observations does not seem all that compelling. It is already known that using non-associative adaptive processes that favour sparse coding, trained with natural images, produces output that resembles complex cell receptive fields. Does this study produce results that are notably more aligned with data from insect lobula recordings, for example?

Line 40 – the "cognitive feats in visual learning" explored in this paper do not seem all that "remarkable".

Lines 52-72 This passage seems to interchangeably use three different senses of 'adaptive': adaptive in the sense of ongoing change in neurons due to the experience of the individual (lines 55-57); adaptive in the sense of being evolutionarily well adapted (lines 57-59); and adaptive in the sense of being versatile and robust (lines 59-61). It would be helpful to keep these differences clear, especially as the claim in this paper is that adaption in the first sense is needed to support adaption in the last sense.

Lines 84 and 92-93 It is not clear why it is stated that sampling "builds up a representation/picture of the environment". Indeed the authors' own work here and previously clearly demonstrates how active sampling can be used to solve visual problems without "building up" a picture.

Line 99 – This is an explicit claim that the paper explores "the necessary and minimally sufficient circuit". However, the paper does not demonstrate necessity or minimality of the circuit elements.

Line 104 – again a claim that the lobula encoding used here is "necessary".

Line 110 – here and later it is claimed the lobula representation is "efficient" but efficiency is never explicitly defined or shown.

Lines 115-116 It seems extremely strange to cite no papers later than 2012 for "neural mechanisms of associative learning in insect brains".

116-117 "visual flight dynamics" in this paper are hugely simplified to a five-step constant speed horizontal scan, so their influence on the model seems overstated here.

121-122 another list of citations going no later than 2013, in an area of very active research.

Line 143 is there "recurrent neural connectivity" between photoreceptors and the lamina (in the model or reality)?

147-148 If I have understood this correctly, the connectivity between medulla and lobula is fixed in advance to be exactly five inputs, arranged in space, and with delay times, to match the standard scanning process used in these experiments. So it assumes the movement of the bee is known? This seems a very arbitrary wiring, is there any evidence to support it? Possibly I have misunderstood but if the spatial extent and timing of these connections is actually created through the non-associative adaptation process, then this has not been well explained in the paper (including the methods).

Figure 1 caption – is it correct that there are random connections (not all-to-one connections) from KCs to the single MBON? Or is the meaning here that the connections have initial random weights? Please clarify.

Figure 2 (and 3) it would be nice to include (where possible) data from actual bee behaviour – how well do they perform relative to the model?

Figure 2 I assume the paired columns in C are similar to those in D, I.e. showing the result if the positive training is to one symbol or the other. If so, it would help to have the same pattern and legend in C.

210-212 I find it mystifying why this circuit should be unable to do the discrimination task when the whole pattern is scanned. Do the lobula neuron responses look the same for both stimuli in this case? Why? Isn't this a significant weakness for the model – that some types of (rather simple) patterns cannot be learned? Frankly, this is much more striking than the fact that the face stimuli can be learned. Please discuss.

Figure 3B the text says the test cases were "a novel grating and a single bar" but the picture appears to show a grating pair that were used in training.

Also, Figure 3, the caption says "except for (A) all simulations were conducted at the default distance …etc." so what was used for A, and why not the default?

Line 241-242 it seems like an overinterpretation of these very mixed results to say "the model was able to extract more than a single feature during its scan of the pattern".

Line 261 and following, there are several claims here that the lobula encoding is efficient. But how is efficiency defined and measured here? Similarly, line 282-284 says the representation is 'decorrelated and sparse' but the only evidence provided seems to be that in example 4B, only a few lobula neurons have high activity.

309-310 If variability between lobula neurons is reduced with fewer neurons, doesn't that argue against the claim that the adaptive process makes them 'decorrelated and sparse'?

See my public review for comments on the claims made in the Discussion that I believe to be insufficiently supported.

*Reviewer #3 (Recommendations for the authors):*

I believe that this very interesting manuscript would benefit greatly from a more in-depth consideration of the terminology and a clearer description of the model and the methods.

I will provide here some more specific points. Below that are the in-line comments.

General notes:

It can be confusing to show the whole pattern when the actual input to the network is only a part of it. A suggestion would be to show, in the graphs, only the actual input to the network.

Should the model architecture used for the results be specified earlier? First mention of number of neurons in lobula is at line 306 (maybe give a name/code to the model variants and refer to those in the method section)

The videos showing the evolution of the receptive fields over the training steps are appreciated, and they could benefit by including a title that describes what is being watched (similar to the caption for images). Possibly, also report the number of training examples over the total of training examples, to show how the receptive fields evolve over time (e.g. Video 5).

Methodology:

In general, the methodology is described somewhat sparsely. Some crucial steps and details are not reported fully, and the full mathematical model of the network is not immediately clear. This is a pity, as it affects the interpretation and may undermine the meaningfulness of the results.

The authors specify that the model considers only green photoreceptors. It is unclear whether this is obtained by processing only the green channel of an original RGB image or by other means.

Lines 488-489. The mathematical notation does not look coherent. Does $A_0$ refer to the $a$ in $f(x; a, b)$? Also, it is not clear whether $A_0$ and the other parameters of the sigmoid activation function ($m$ and $b$) have fixed values or are parametrized and learned. The reviewer assumes they are fixed. Finally, it can be slightly confusing what does $r_p$ represent in the equation. I assume it represents the activity of one green photoreceptor $p$, and that $P$ represents the total amount of photoreceptors (pixels in the image) considered as input to one lamina neuron (as such, P=9).

Line 486-489. The tiling of the receptive field of each Lamina unit is not specified. Given the reported numbers (of pixels and units), the reviewer assumes that the tiling is formed by 625 squares of 3x3 pixels, each adjacent but non-overlapping with the others.

It is not written how the output of each photoreceptor ($r_p$) is obtained from the input image, nor whether it is a continuous or discrete value.

The superscripts are never mentioned explicitly, and the reader is left to infer that they refer to the different components of the network architecture (e.g., $La$ = Lamina). Albeit not critical, this could become an issue when considering that other parts of the mathematical notation are also not detailed, leaving possibly too much room for interpretation.

Line 500-501. What is the parameter $\λ$ of the Poisson distribution used to generate the noise in the activation?

The topology of connection from Lamina units to Medulla units is unclear to this reviewer. Lines 505-507 specify that Medulla units have a small (receptive) field, each one being activated by a different region of the image patch. From Figure 1B, the image patch seems to be one full 75x75 frame. The manuscript, however, does not report what is the small field (selected from this patch) to which one Medulla unit is said to respond. The total number of Medulla units also seems not to be reported.

The topology of connections from Medulla units to Lobula units is unclear to this reviewer. Lines 503-505 state that a total of M Medulla units is connected to each Lobula unit. However, the Methods do not describe how these M units are selected from those in the Medulla. Are they adjacent to each other? If so, in what order is the temporal delay applied?

It is not clear whether, when observing half or a corner of a pattern, the amount of Medulla units changes to reflect a lower number of pixels in the image, or the original image is enlarged to keep the current network configuration, changing the scale of the observed features, or neither of the above. The lack of clarity about the topology of connections from Lamina to Medulla and from Medulla to Lobula makes it difficult to interpret what happens in this case.

It is not reported from which distribution is the random connectivity matrix S initialized, nor whether it is randomly reinitialized for each simulated bee.

It is not reported how the laterally inhibitory connections in the lobula, Q, mathematically affects the neurons activity.

It is not reported over which window of time is the mean firing rate computed, nor if it computed as a static condition of the leaky integrate-and-fire model assuming a fixed value of activity in the input layer (Medulla) during the current training step.

When the receptive fields are shown in figures, the weights seem to be clipped in the range [-1, +1]. However, no clipping is reported in the Methods.

A formal description of all the initialization, training, and testing steps is not reported, and it's left to be inferred from different parts of the manuscript.

Discussion

The Discussion section of this work is somewhat lacking when it comes to analyzing the variation of performance of the proposed network over the whole spectrum of tested conditions.

It is the belief of this reviewer that the underperformance of the network in those cases, and with some type of patterns (even in the best-performing scenario, such as with the gratings in Figure 3D), could be attributed to the receptive fields that are formed during the non-associative learning procedure. Specifically, the receptive fields shown seem to all be responsive to a specific orientation and velocity. However, they are all "global" (or "large" scale), in the sense that they all have only one big contiguous area of positive weights along one big contiguous area of negative weights.

The question that naturally arises is whether the non-associative learning employed here can produce more refined patterns in the receptive fields, or whether it can learn to be sensitive over different scales of features by combining (at the level of the lobula) different large-scale receptive fields in non-trivial ways. Also, why do patterns learned, when scanning whole images, perform best when not applied to a whole image.

The reviewer acknowledges the difficulty in showing the receptive fields when they include both a spatial and temporal component, and as such also that this sensitivity on a varying scale could already be present in the network, as a combination of larger scale receptive fields (albeit the results with different speeds and different distances from the image seems to suggest some specificity in the scale of visual feature that the network can identify). This could warrant a more in-depth study of the performance of the proposed network architecture when varying the training set and training conditions (e.g., speed).

In-Line comments

L134-136: but is the scanning order selected by the model, or is it fixed? At the moment it seems to be implied that the 5 frames are given. Is there reason to assume this is the optimal order? Is there any optimal order? P.S. I see that this is touched upon below. See comment to L210-219

L185-188: Videos 1 and 2 are missing. I assume they are present in MaBouDi et al. 2021, but if they are referenced here with this indication they should be included. Alternatively, you could add in the text a general reference to the previous paper.

L196-197: writing here "during the initial experiment" seems to refer to the first experiment you are doing in this paper. Instead, it seems you want to refer to your previous paper. This should be made more clear.

Figure 2B – In the caption, there seem to be typos on what is rewarding and what is punishing.

200-201 – This wording may be interpreted as the model having active control, which is however not the case.

L201-219: statistical analysis should be done and reported. In an attempt to comparing this model with living animals, I believe every step should be taken to follow the same procedure. How many simulated bees have been tested in the + vs X task? Are they 20 per shape or 20 per scanning pattern? Is the data collected after how many visits? Reporting percentage is I believe insufficient, and a binomial test against chance level should be performed. This is the case for all experiments in this paper.

Figure 2B: is this SD or CI?

L210-219: the authors here refer to an initial poor performance. In figure 2C, is this the last two bars of the graph? This should be made clear.

Overall, the experiments here described aimed at finding the best scanning procedure, but I am not sure if how this was evaluated is appropriate. First, how is the training on natural images organized? Are they all scanned in the bottom half, left to right? If that is the case, the highest efficiency of lower-half scanning may be linked to the highest similarity to training, not to the real efficiency of the technique, and it would as such suggest low generalizability of the model. If instead training is repeated for all experiments following that presentation pattern, and thus the scanning procedure has an effect of learning effectiveness, this may be a property of this network, but not necessarily of living bees (as L216 somewhat suggests). It could in fact be a self-fulfilling prophecy (behavioral experiments suggest the need for scanning, the network is designed with a recurrent layer to enable shape reconstruction, and the network is most effective with scanning).

If you want to suggest that scanning from the bottom left is indeed more effective, you need to also include conditions other than the confirmatory one. These could be scanning right to left, or scanning left to right in the top half, or scanning top to bottom, or even diagonally (which I suspect are going to produce identical results). As of now, the experimental conditions only allow us to conclude that scanning sections is more effective than seeing the whole image, which again is to me included as a property of the network. Also, I may be wrong about this, but bees visual field is not centered frontally on the animal, but points upward (https://www.researchgate.net/publication/326717773_Bumblebee_visual_allometry_results_in_locally_improved_resolution_and_globally_improved_sensitivity). Being this the case, a bee moving across the bottom of a stimulus wouldn't it actually be looking at it fully, with the visual field centered on the horizontal symmetry line?

A similar reasoning should be made for scanning speed. Velocity is tangled with stimulus size. 0.1m/s may work best with this size but will change drastically depending on how much of the stimulus occupies the virtual bee visual field.

I want to point out that none of these points are detrimental to the effectiveness of the model itself, which seems to present good performances. But if claims want to be made about the best scanning strategy, especially if confronted with real animals, these points should be tested and addressed. As of now, we can say that the current model best performs under certain conditions, but we can't generalize the effectiveness of such conditions to be the best for the task, nor the best for bees.

L221-252: I believe also here binomial statistics should be produced. I understand it seems to be redundant for performances nearing 100%, but this becomes more relevant for the 60% and 40% reported in Fig3E. On the same note, specific values should be reported, both for averages and SD.

246-251 – Repetition of a period.

It would be helpful if Figure 3 also reported the real bees data, as taken from the various papers. This would give a sense of how closely the model follows the bees behavior. Of course, bees are more complex and are subject to, among others, motivational effects which will make the choice percentage less clean, but I still think this would be appreciable.

311-312 – "When the model is limited to only four neurons in the lobula, it lacks the capability to encode the entire spatio-temporal structure that is naturally present in the training patterns". This wording seems to suggest that with more neurons it can encode the entire spatio-temporal structure of the training patterns, which may be an overstatement.

L314-316: I agree that these neurons are sufficient for the discrimination task in hand, but I am unsure whether is appropriate to extend this to bees, as the paragraph title implies. Bees have to use the system to respond to much more complex patterns, like photorealistic ones. For example, is 16 neurons still enough for the face discrimination task?

355-356 – It is unclear how the study would suggest a crucial role of movement in the ability to efficiently analyze and encode the environment. In this work, movement of the input pattern is taken as a given condition under which the network is trained, and not as a tool that can be exploited to have an advantage in the encoding and analysis of the pattern itself.

Discussion: In general, I am not fully convinced that your model can say anything about the bees or the optimal performance in general, but should focus on the effectiveness of the model itself. This is because of what I have reported above about how the model performance is at least partially dependent on the model design, and not on how bees actually behave (which is hypothesized)

374-375 – In these lines, it is claimed that the model acts as a linear generative model, however, this is not shown in the results and these generative capabilities are not demonstrated.

487 – Calling $r_l^{La}$ as "the output of one lamina neuron" instead of "one lamina neuron" could improve clarity.

498 – I have not clear what "however" refers to, in this context

498 – Similarly. Rewording "the input of the m −the medulla neuron is calculated" to something like "the input to the $m$-th medulla neuron, $I_m{Me}$, is calculated as"

506-507 – Could reference to Figure 3B be a typo?

528-530 "At each step of training, a set of five patches with size 75x75 pixels, selected by shifting 15 pixels over the image from the left or right or the reverse orientation (Figures1B, 2A), was considered as the input of the model." This wording could be a bit confusing, especially as, coincidentally, 15*5=75. It could be improved to make it clear that one input to the network is (a concatenation?) of 5 patches of 75x75 pixels each, obtained by shifting a window of 75x75 pixels by 15 pixels, 5 times (if this is actually the case).

*Reviewer #1 (Recommendations for the authors):*

The revised manuscript incorporates my comments well. The added analysis better clarified that local visual features are essential for learning using this scanning strategy. The description was also significantly revised, and the claim sounds reasonable now. I do not have further comments.

---

## [Author Response]

Essential Revisions:The reviewers agree that the model developed by the authors tackles an important problem in machine learning and visual neuroscience. The model is based on visual scanning, which represents a novel and exciting phenomenology in the bee. Unfortunately, the main conclusions of the work must be watered down until the authors demonstrate that alternative, equally plausible, models of the visual and mushroom body circuits are not sufficient to solve the tasks under consideration. We believe that the manuscript can be a valuable and important contribution to the field if the following weaknesses are thoroughly addressed.

Thank you for compiling this detailed list of essential revisions. We appreciate it, as well as the recognition of the importance and novelty of our study. In response to the outlined revisions, we have substantially improved the manuscript by incorporating additional analyses, control models, deeper discussions, and methodological clarifications. Below, we summarise the key revisions and how they comprehensively address each of the essential concerns.

1) The neural circuitry underlying the model does not adequately integrate the wealth of *Drosophila* connectome data that has been published during the past 10 years. While the model is definitely bio-inspired, layers of its architecture are built very differently from the connectivity of real insect brains. As a result, many features of the model's architecture appear to be arbitrary. Clarifications should be provided about circuit-function relationships of the bee MB versus the Drosophila MB, and their implications on the model.

We acknowledge that our original manuscript did not fully incorporate recent *Drosophila* connectome data into the model’s design. However, this was an intentional simplification aimed at identifying core computational principles underlying visual learning, rather than precisely replicating the anatomical complexity of the insect brain. Below, we outline our reasoning and the broader implications of our approach.

Simplification for Computational Insight

– A key objective of our study was to uncover the fundamental computational mechanisms underpinning active vision and visual pattern recognition in insects, with a particular focus on bees, as they are the only insect species that has been experimentally tested for active vision principles in visual pattern recognition, which is the central question of this study.

– Incorporating highly detailed connectome data would introduce a large number of additional parameters, making it difficult to isolate and interpret the contributions of individual components.

– By abstracting certain details, we can systematically assess how different computational principles—such as spatiotemporal encoding, sequential scanning, and non/associative learning—contribute to pattern recognition.

II.Limitations of Connectome Data for Functional Understanding

– While *Drosophila* connectome data provide an unprecedented level of structural detail, they do not directly reveal functional relationships or behavioural significance, although they have the potential to contribute to understanding these aspects.

– The functional role of specific circuit motifs in the mushroom body (MB) for visual pattern recognition remains largely untested; most of the available information for fruit flies relates to olfactory learning.

– Our model serves as a predictive tool, offering insights into how motion and spatiotemporal encoding contribute to visual learning—hypotheses that neurobiologists can test using connectomic and physiological approaches.

III.Importance of Motion in Visual Processing

– Most neuroscientific studies on insect vision have focused on either navigation or motion coding, but very few have directly addressed the neural correlates of pattern recognition.

– Our work highlights the importance of motion in visual recognition, suggesting that sequential visual input (as seen in active vision) fundamentally shapes neural encoding.

– This is particularly relevant because most visual studies in *Drosophila* have relied on static viewing condition, which do not account for the role of motion in shaping neural responses.

IV.Why We Focused on Bee Data

– Unlike *Drosophila*, bees have a longer history of research and a larger body of data on behavioural and neural aspects of visual learning and pattern recognition.

– While some level of abstraction is intended, our model does draw from established electrophysiological, computational, and behavioural studies in bees, aligning with known experimental findings in visual learning tasks.

– While connectome data from *Drosophila* provide valuable structural insights, there is no direct connectomic study in flies that tackles the neural correlates of pattern recognition.

In summary, we acknowledge the significance of connectome data in advancing our understanding of neural circuits; however, our approach prioritises computational abstraction to uncover the core principles of active vision and visual learning. Rather than fully replicating anatomical details, our model provides testable hypotheses that can inform future neurobiological investigations in bees, *Drosophila*, and other insect models. By emphasising motion-based visual processing, our study broadens the scope of insect vision research and highlights the importance of dynamic visual input—a factor often overlooked in traditional static-image studies. We believe this computational perspective can inspire new experimental approaches that integrate both static and dynamic visual cues in behavioural and neurophysiological studies. The simplifications inherent in our model have been thoroughly discussed in multiple sections of the Discussion, highlighted in blue, where we outline their implications and limitations while justifying our design choices.

2) Given the repeated claims that the authors present the "minimal circuit" required for the visual tasks explored, the work ought to rigorously and systematically assess the necessity and sufficiency of the different components included in the circuitry of the model. In particular, could a simpler learning rule be sufficient to explain discrimination? In what sense is the presented circuit "minimal"? Varying the number of lobula neurons of the model is a good first step, but the same should be done for other components of the model.

The reviewers rightly pointed out that our previous claim of a "minimal circuit" was not sufficiently justified. In response, we have significantly refined this aspect in the revised manuscript to ensure that our conclusions are well-supported and appropriately cautious.

Our primary aim was not to define an optimally minimised model, but rather to identify a functional computational circuit that enables visual coding and learning. Specifically, we sought to explore the role of key model components, such as non-associative learning, natural image statistics, the number of neurons involved in active vision, and scanning behaviour. This study represents an initial step in a novel approach, and we acknowledge that further refinements and investigations will be necessary. We have explicitly discussed the limitations of our model in Discussion section and made significant revisions regarding this claim, as summarised below:

Revisions and Improvements:

– We have removed overstatements about minimality and now clarify that our model represents one possible functional circuit rather than the absolute minimal configuration.

– To rigorously assess necessity and sufficiency, we conducted new control experiments:

– We revised the relevant sections to examine the influence of each model component on the final performance, rather than focusing solely on the number of lobula neurons.

– The revised manuscript includes a detailed justification for why specific mechanisms (e.g., spatiotemporal encoding, lateral inhibition, neuromodulation interactions) are necessary for task performance.

These revisions significantly strengthen the evidence for our model’s functional relevance while ensuring that our claims about minimality are well-contextualised and appropriately framed. Furthermore, we provide a clear rationale for our intentional simplifications in designing the network, allowing for a more interpretable assessment of computational principles in active vision.

3) The presentation of model's results and its interpretation should explain the successes and failures of the model in reproducing the actual behavior. It should include a more in-depth comparison of the performances of the model and real bees.

We apologise that the previous submission did not provide sufficient clarity on the conditions under which the model succeeds or fails. In response, the revised manuscript now presents a more comprehensive and in-depth analysis of performance variability, ensuring a clearer comparison between model outcomes and real bee behaviour. The following key revisions and improvements have been made:

– We have conducted a statistical comparison between the model's performance and compare them with real bee behaviour across multiple tasks, providing a more quantitative and systematic evaluation.

– The Results and Discussion sections have been substantially expanded to explicitly examine:

– Why certain complex patterns (e.g., spirals, faces) are successfully learned, whereas others (e.g., plus vs. multiplication symbols) can fail under specific scanning conditions. We have performed control experiments and deeper analysis to investigate the underlying mechanisms of success and failure, ensuring that these outcomes are well-explained. Additionally, further experiments highlight the conditions under which the current stage of the model succeeds or fails.

– How scanning strategies influence encoding and generalisation, drawing direct comparisons with bee vision behaviour to assess whether the observed results align with biological findings.

– The impact of scanning trajectory constraints, distinguishing between limitations imposed by model design and true behavioural optimisation strategies in bees.

– Figure 3 and Figure 5 have been revised to better illustrate model failures, providing clearer explanations of why specific conditions lead to reduced performance.

These revisions enhance the interpretability of the model results, ensuring that both successes and limitations are well-contextualised within their biological relevance.

4) The description of the methodology is incomplete, which prevents a proper interpretation of the model's results.

We apologise for the lack of methodological details in the previous submission, which may have limited the readers’ ability to fully interpret our results.

In response to the reviewers’ suggestions, we have substantially expanded the Methods section, providing a clearer and more comprehensive breakdown of:

– Network architecture, connectivity, and parameter choices, ensuring a detailed description of the model's structure.

– Training and testing procedures, incorporating new control experiments to further validate our findings.

– Training with natural images and its impact on receptive field formation, offering deeper insights into how the model learns visual patterns.

– Details on scanning sequences and how they were integrated into the model, improving transparency in the experimental design.

These revisions significantly enhance methodological clarity, ensuring that readers can better understand, evaluate, and reproduce our approach.

Reviewer #1 (Recommendations for the authors):The introduction talks about natural scenes such that their specific features are critical to neural processing and pattern recognition. However, the manuscript does not thoroughly assess if the model is particularly suited for handling natural scenes. To demonstrate this, it appears to require using non-natural images for non-associative learning and comparing the performance with the model trained with natural scenes. Otherwise, I would recommend rephrasing the introduction.

We appreciate your insightful comment regarding the need to assess whether the model is particularly suited for handling natural scenes. To address this, we trained the model using shuffled natural images, in which the spatial structure was disrupted while preserving low-level statistical properties such as pixel values. This approach allowed us to isolate the effect of natural image structure on the model's performance. Our results demonstrate that the model trained on shuffled natural images failed to develop any structured receptive fields—its connectome remained random—resulting in a complete lack of discriminability and sparseness (Figure 3, Figure 2-suplementary figure 1B). This finding indicates that the specific spatial structure of natural scenes plays a crucial role in shaping the model’s learning and recognition capabilities. These results further support the premise introduced in the manuscript, reinforcing the argument that neural processing is particularly attuned to the statistical properties of natural scenes. We have now expanded this discussion in the manuscript, dedicating a full paragraph in the Discussion section and including an additional figure to illustrate these findings.

Line 217: why does restricting the scan field improve the performance dramatically? This may be intuitively reasonable, but it would be nice to have an explicit explanation based on the model's structure.

We apologise for the lack of clarity in our previous explanation. The restriction of the scan field significantly enhances performance by providing a more focused and high-contrast input, allowing for clearer and more relevant visual information to drive strong activation in lobula neurons. In contrast, when the entire image is processed, the response becomes less sparse, resulting in broader but weaker activation across the lobula. This diffused activation reduces stimulus discriminability in the subsequent associative layer, which relies on sparse and selective responses for effective differentiation.

We have now incorporated this explanation into the manuscript, expanding the discussion on the model’s structural dynamics and how scan restriction influences neural encoding and pattern recognition (Figure 5, 6F). Furthermore, our model predicts that bees may focus their scanning on smaller regions when confronted with more complex patterns, allowing them to extract simpler, more localised information. This aligns with experimental findings on bee visual behaviour (MaBouDi et al., 2021). For further clarification, please refer to our response to the Journal above.

Discussion:The insect lobula is not necessarily only composed of wide-field neurons. It would be nice to have some discussion about how other types of neurons, such as small object-detecting neurons, could contribute to the same visual task.

Thank you for your suggestion. The involvement of other cell types in visual processing, beyond achromatic pattern recognition, has been discussed in the main text in lines 544- 558.

Reviewer #2 (Recommendations for the authors):Line 1 – title – is it really substantiated in this paper that the spatiotemporal lobula encoding "aids" pattern recognition? Relative to what? Can these tasks not be solved by models such as that in Ardin et al. (2016) that use low resolution pixel values as input to the KC and associate the corresponding sparse code with the MBON for selected images?

Our study does not claim that spatiotemporal encoding is the only possible mechanism for pattern recognition; rather, we investigate how motion-driven visual processing and sequential scanning strategies influence neural coding and pattern recognition. Unlike models such as Ardin et al. (2016), which use low-resolution pixel values as direct input to Kenyon cells (KCs), our model incorporates a biologically inspired intermediate processing stage in the lobula, where visual patterns are dynamically encoded through temporal integration and lateral inhibition. This results in a sparse, decorrelated representation that aligns with known functional properties of insect visual neurons.

While simpler models, such as Ardin et al. (2016), may also solve specific pattern recognition tasks, they do not account for the role of active vision and require computationally expensive processing of all pixel values directly within the mushroom body. Our findings suggest that the visual system, particularly encoding within the lobula and lobula plate, generates a sparse, uncorrelated, and selective input to the learning centre, such as the MB. This compact, feature-selective representation reduces computational complexity, as only a small subset of highly selective neurons needs to be processed in the mushroom body and associated with reinforcement signals. Such an encoding strategy is not only more efficient but also more robust to input noise, potentially enhancing generalisability compared to direct pixel-wise input models.

Additionally, we discuss the broader implications of our findings on lines 560-578, particularly in visual navigation, by comparing the advantages of our approach to models like Ardin et al. (2016), which process raw pixel values without leveraging the benefits of motion-driven encoding and structured visual input filtering. These insights provide a more biologically plausible mechanism for efficient visual processing in insects while also offering potential applications for bio-inspired artificial vision systems.

Line 6 – abstract – in what sense is the presented circuit 'minimal'? The paper explores reducing the number of lobula neurons, but not any other reduction in complexity.

The abstract has been revised.

Line 11 – the alignment to neurobiological observations does not seem all that compelling. It is already known that using non-associative adaptive processes that favour sparse coding, trained with natural images, produces output that resembles complex cell receptive fields. Does this study produce results that are notably more aligned with data from insect lobula recordings, for example?

There is currently no population-level data on lobula neuron activity that would allow for a direct comparison of our model’s predictions on sparsity with empirical data. However, our model’s results were compared to functional recordings of lobula neurons involved in pattern recognition (Figure 2), demonstrating alignment with observed response properties. Given that early visual processing systems are widely recognised to support efficient coding by producing sparse and decorrelated activity, our findings are consistent with established neurobiological principles in primates. This has been further discussed in the Discussion section (lines 524-535).

Line 40 – the "cognitive feats in visual learning" explored in this paper do not seem all that "remarkable".

Numerous studies have highlighted the remarkable visual learning abilities of bees, particularly given their small brain size compared to other animals. While face recognition is typically considered a complex cognitive task, our model—when aligned with previous bee experiments—demonstrates that bees can learn and discriminate human faces, a challenge that has been explored in larger-brained animals. This supports the view that even miniature neural circuits can achieve sophisticated visual discrimination.

Lines 52-72 This passage seems to interchangeably use three different senses of 'adaptive': adaptive in the sense of ongoing change in neurons due to the experience of the individual (lines 55-57); adaptive in the sense of being evolutionarily well adapted (lines 57-59); and adaptive in the sense of being versatile and robust (lines 59-61). It would be helpful to keep these differences clear, especially as the claim in this paper is that adaption in the first sense is needed to support adaption in the last sense.

Thank you for pointing this out. This paragraph has been revised to clarify the distinction between evolutionary adaptation and experience-dependent adaptation. Please see lines 64-89 for the updated version.

Lines 84 and 92-93 It is not clear why it is stated that sampling "builds up a representation/picture of the environment". Indeed the authors' own work here and previously clearly demonstrates how active sampling can be used to solve visual problems without "building up" a picture.

This has been revised to "build up a neural representation of their environment."

Line 99 – This is an explicit claim that the paper explores "the necessary and minimally sufficient circuit". However, the paper does not demonstrate necessity or minimality of the circuit elements.

This has been revised to ‘Building on our previous work analysing bee flight paths during a simple visual task (MaBouDi et al., 2021b), we further investigated the main circuit elements that contribute to active vision in achromatic pattern recognition, focusing on a simplified yet biologically plausible model.’

Line 104 – again a claim that the lobula encoding used here is "necessary".

The sentence has been revised to ‘We hypothesised that the bees’ scanning behaviours have co-adapted to sample complex visual features in a way that efficiently encodes them into spatiotemporal patterns of activity in the lobula neurons, facilitating distinct and specific representations that support learning in the bees’ compact brain.’

Line 110 – here and later it is claimed the lobula representation is "efficient" but efficiency is never explicitly defined or shown.

We apologise for the lack of clarity and have now revised the manuscript to provide a more precise and detailed explanation of this aspect. The efficiency of our lobula responses, in terms of sparsity, decorrelation, and selectivity, was further analysed and discussed in Figures 3 and 5 (new figures in the revised manuscript). Our results demonstrate that non-associative learning and its related components contribute to an optimised neural representation, reducing redundancy while preserving critical visual information. This aligns with principles of efficient coding, where neuronal activity is sparse and minimally correlated, allowing for a more compact yet informative representation of the environment.

Lines 115-116 It seems extremely strange to cite no papers later than 2012 for "neural mechanisms of associative learning in insect brains".

We acknowledge the reviewer's concern. In this section, we specifically reference earlier papers that directly informed our model development. However, in the Discussion, we refer to more recent studies when discussing the plasticity rules in the mushroom bodies, providing a broader context for associative learning mechanisms in insect brains.

116-117 "visual flight dynamics" in this paper are hugely simplified to a five-step constant speed horizontal scan, so their influence on the model seems overstated here.

The limitations of our model have been addressed and highlighted in various paragraphs of the Discussion, particularly regarding the simplifications in visual flight dynamics. Additionally, we have outlined future directions, including the implementation of a more comprehensive visual flight dynamics model to further enhance our understanding of active vision.

121-122 another list of citations going no later than 2013, in an area of very active research.

The references have been updated to include recent research.

Line 143 is there "recurrent neural connectivity" between photoreceptors and the lamina (in the model or reality)?

This paragraph describes the model structure. However, it has been revised to improve clarity and reduce potential misunderstandings.

147-148 If I have understood this correctly, the connectivity between medulla and lobula is fixed in advance to be exactly five inputs, arranged in space, and with delay times, to match the standard scanning process used in these experiments. So it assumes the movement of the bee is known? This seems a very arbitrary wiring, is there any evidence to support it? Possibly I have misunderstood but if the spatial extent and timing of these connections is actually created through the non-associative adaptation process, then this has not been well explained in the paper (including the methods).

Apologies for the lack of clarity. Additional details have been added to the beginning of the result section and Methods section along with a paragraph discussing the potential mechanisms underlying this proposed network. The choice of five inputs is a model simplification to represent the scanning input to lobula neurons rather than an exact biological wiring. The model assumes prior knowledge of the bee’s speed and simplifies the structured visual input based on a part of observed scanning behaviour. The rationale behind this, as well as the optimisation and order of these inputs, is further discussed.

Figure 1 caption – is it correct that there are random connections (not all-to-one connections) from KCs to the single MBON? Or is the meaning here that the connections have initial random weights? Please clarify.

For each simulation, the model is set with a randomly weighted connectivity matrix, S. This has been revised in the figure caption and further explained in Methods.

Figure 2 (and 3) it would be nice to include (where possible) data from actual bee behaviour – how well do they perform relative to the model?

Following the advice of other reviewer, we applied statistical tests to the data and updated the figures to indicate significance with ‘*’ for statistically significant differences and ‘n.s.’ for non-significant results. The details of the statistical analysis have also been provided in the Methods section on lines 875- 893.

Figure 2 I assume the paired columns in C are similar to those in D, I.e. showing the result if the positive training is to one symbol or the other. If so, it would help to have the same pattern and legend in C.

Thank you for your suggestion. Panel C has now been updated and is presented as Panel D in the revised Figure 4.

210-212 I find it mystifying why this circuit should be unable to do the discrimination task when the whole pattern is scanned. Do the lobula neuron responses look the same for both stimuli in this case? Why? Isn't this a significant weakness for the model – that some types of (rather simple) patterns cannot be learned? Frankly, this is much more striking than the fact that the face stimuli can be learned. Please discuss.

Apologies for the lack of interpretation regarding the model’s performance. The lobula neuron activity has now been analysed in detail and is presented in Figures 3, 5, with further discussed in the result and Discussion section. In summary, when the entire pattern is scanned, a larger number of lobula neurons are activated; however, their responses are weaker and less selective, failing to provide a strong enough input for Kenyon cells to effectively associate the stimulus with reward or punishment, leading to poor discrimination performance.

This finding aligns with a key aspect of bee movement—bees may need to approach stimuli closely for effective scanning rather than making decisions from a distance (see MaBouDi et al., 2021). Close-range scanning enhances feature selectivity and learning, whereas scanning an entire pattern from a distance lead to diffuse, less discriminative activation, reducing the effectiveness of pattern differentiation. We have now explicitly addressed this interpretation and its implications in the Discussion section.

Figure 3B the text says the test cases were "a novel grating and a single bar" but the picture appears to show a grating pair that were used in training.

The model was tested on a novel pair of gratings with a single bar, different from those used in training.

Also, Figure 3, the caption says "except for (A) all simulations were conducted at the default distance …etc." so what was used for A, and why not the default?

This has been corrected to: "All simulations were conducted at a fixed distance of 2 cm and a scanning speed of 0.1 m/s, focusing on the lower half of the pattern."

Line 241-242 it seems like an overinterpretation of these very mixed results to say "the model was able to extract more than a single feature during its scan of the pattern".

This has been revised.

Line 261 and following, there are several claims here that the lobula encoding is efficient. But how is efficiency defined and measured here? Similarly, line 282-284 says the representation is 'decorrelated and sparse' but the only evidence provided seems to be that in example 4B, only a few lobula neurons have high activity.

We have conducted further analysis to provide a clearer definition and quantification of efficiency, sparsity, and decorrelation in the model’s lobula encoding. Specifically, we now define efficiency in terms of sparse coding and reduced redundancy in neural responses, which enhances the selectivity and discriminability of visual features.

To support these claims, we have included additional quantitative measures, such as sparsity indices and correlation analyses, demonstrating how lobula neurons develop decorrelated and selective responses to visual stimuli. These results are now explicitly discussed in Figures 3 and 5, as well as in the revised text.

309-310 If variability between lobula neurons is reduced with fewer neurons, doesn't that argue against the claim that the adaptive process makes them 'decorrelated and sparse'?

We observed that reducing the number of lobula neurons may lead to lower variability in their responses, which could seem to contradict the claim that the adaptive process promotes decorrelation and sparsity. However, the relationship between neuron count, decorrelation, and sparsity is not strictly linear.

Our analysis shows that sparsity and decorrelation emerge as a function of competition among lobula neurons, which is driven by non-associative learning and lateral inhibition. When the number of lobula neurons is reduced, the diversity of receptive fields also decreases (Figures 2, 8), which limits the range of features that can be selectively encoded. This results in less decorrelation at the population level, as fewer neurons are available to distribute feature encoding efficiently.

Reviewer #3 (Recommendations for the authors):I believe that this very interesting manuscript would benefit greatly from a more in-depth consideration of the terminology and a clearer description of the model and the methods.I will provide here some more specific points. Below that are the in-line comments.

We appreciate your positive feedback and constructive suggestions. In response, we have carefully revised the manuscript to enhance clarity in both terminology and methodology. Specifically, we have refined key definitions to ensure consistency and precision, and we have expanded the model description to provide a more comprehensive and transparent account of its structure and underlying assumptions. We believe these revisions significantly strengthen the manuscript and greatly appreciate your valuable input. Please find our detailed responses below.

General notes:It can be confusing to show the whole pattern when the actual input to the network is only a part of it. A suggestion would be to show, in the graphs, only the actual input to the network.

See answer below, please.

Should the model architecture used for the results be specified earlier? First mention of number of neurons in lobula is at line 306 (maybe give a name/code to the model variants and refer to those in the method section)

Thank you for your suggestion. The structure of the Results section has been revised, beginning with a summary of the network topology and an interpretation of the result of non-associative learning to provide greater clarity and coherence for the reader.

The videos showing the evolution of the receptive fields over the training steps are appreciated, and they could benefit by including a title that describes what is being watched (similar to the caption for images). Possibly, also report the number of training examples over the total of training examples, to show how the receptive fields evolve over time (e.g. Video 5).

Thank you for your suggestion. We have expanded the figure captions to provide clearer and more detailed information for the audience.

Methodology:In general, the methodology is described somewhat sparsely. Some crucial steps and details are not reported fully, and the full mathematical model of the network is not immediately clear. This is a pity, as it affects the interpretation and may undermine the meaningfulness of the results.

We apologise for not providing sufficient details about the model in the previous version. The updated manuscript now includes more comprehensive descriptions and additional details.

The authors specify that the model considers only green photoreceptors. It is unclear whether this is obtained by processing only the green channel of an original RGB image or by other means.

Since we aim to compare the model’s output with bee behavioural data and the model is not designed for colour learning, we restricted it to only green photoreceptors. This decision is based on the hypothesis that bee pattern recognition primarily relies on the green component of visual input. Among the three types of photoreceptors in honeybees, green-sensitive photoreceptors are the most numerous, making them dominant in the bee's vision system.

Additionally, natural images exhibit a strong correlation between the red, green, and blue pixel values. Given this correlation, processing only the green channel should not significantly impact the final result. We have now clarified this point in the manuscript on lines 659-671.

Lines 488-489. The mathematical notation does not look coherent. Does $A_0$ refer to the $a$ in $f(x; a, b)$? Also, it is not clear whether $A_0$ and the other parameters of the sigmoid activation function ($m$ and $b$) have fixed values or are parametrized and learned. The reviewer assumes they are fixed. Finally, it can be slightly confusing what does $r_p$ represent in the equation. I assume it represents the activity of one green photoreceptor $p$, and that $P$ represents the total amount of photoreceptors (pixels in the image) considered as input to one lamina neuron (as such, P=9).

Sorry for the lack of clarity. We have now corrected this and provided a more detailed explanation in the revised manuscript.

Line 486-489. The tiling of the receptive field of each Lamina unit is not specified. Given the reported numbers (of pixels and units), the reviewer assumes that the tiling is formed by 625 squares of 3x3 pixels, each adjacent but non-overlapping with the others.

We have now clarified the tiling of the receptive fields. Each lamina neuron receives input from a non-overlapping 3×3 grid of neighbouring photoreceptors, ensuring full coverage of the visual field without redundancy. This structured arrangement has been explicitly stated in the revised manuscript.

It is not written how the output of each photoreceptor ($r_p$) is obtained from the input image, nor whether it is a continuous or discrete value.

Each photoreceptor’s output is computed directly from the pixel intensity of the corresponding location in the input image, then normalized between 0 and 1, ensuring a smooth response that reflects the natural variation in luminance. This clarification has been added to the revised text.

The superscripts are never mentioned explicitly, and the reader is left to infer that they refer to the different components of the network architecture (e.g., $La$ = Lamina). Albeit not critical, this could become an issue when considering that other parts of the mathematical notation are also not detailed, leaving possibly too much room for interpretation.

Thank you for your suggestion. We have now explicitly defined all superscripts in the manuscript, ensuring clarity in their reference to different network components.

Line 500-501. What is the parameter $\λ$ of the Poisson distribution used to generate the noise in the activation?

Additional details have been added to the revised manuscript for further clarification.

The topology of connection from Lamina units to Medulla units is unclear to this reviewer. Lines 505-507 specify that Medulla units have a small (receptive) field, each one being activated by a different region of the image patch. From Figure 1B, the image patch seems to be one full 75x75 frame. The manuscript, however, does not report what is the small field (selected from this patch) to which one Medulla unit is said to respond. The total number of Medulla units also seems not to be reported.The topology of connections from Medulla units to Lobula units is unclear to this reviewer. Lines 503-505 state that a total of M Medulla units is connected to each Lobula unit. However, the Methods do not describe how these M units are selected from those in the Medulla. Are they adjacent to each other? If so, in what order is the temporal delay applied?It is not clear whether, when observing half or a corner of a pattern, the amount of Medulla units changes to reflect a lower number of pixels in the image, or the original image is enlarged to keep the current network configuration, changing the scale of the observed features, or neither of the above. The lack of clarity about the topology of connections from Lamina to Medulla and from Medulla to Lobula makes it difficult to interpret what happens in this case.

We apologise for the lack of clarity regarding the network topology. The relevant sections in both the Results and Methods have been substantially revised, providing greater detail and clarity. We believe the updated version offers a more comprehensive explanation for the reader.

It is not reported from which distribution is the random connectivity matrix S initialized, nor whether it is randomly reinitialized for each simulated bee.

Apologies for the ambiguity. Additional details have been added to the revised for further clarification.

It is not reported how the laterally inhibitory connections in the lobula, Q, mathematically affects the neurons activity.

Thank you for your suggestion. This now has been discussed in the revised manuscript.

It is not reported over which window of time is the mean firing rate computed, nor if it computed as a static condition of the leaky integrate-and-fire model assuming a fixed value of activity in the input layer (Medulla) during the current training step.

This is now corrected, as the firing rates are already reported in spikes per second.

When the receptive fields are shown in figures, the weights seem to be clipped in the range [-1, +1]. However, no clipping is reported in the Methods.

In the training process, we restricted the synaptic weights to this specific range to facilitate faster convergence. Additionally, this constraint aligns with biological evidence suggesting limitations in synaptic transmission strength, preventing excessive potentiation or depression. We have now clarified this in the Methods section to ensure consistency with the figures presented.

A formal description of all the initialization, training, and testing steps is not reported, and it's left to be inferred from different parts of the manuscript.

Further details have now been included in the updated version.

DiscussionThe Discussion section of this work is somewhat lacking when it comes to analyzing the variation of performance of the proposed network over the whole spectrum of tested conditions.

Further discussion has been added to the Discussion section to interpret and analyse the model’s performance and compare it with other studies. Additionally, please refer to our response to other comments for further details.

It is the belief of this reviewer that the underperformance of the network in those cases, and with some type of patterns (even in the best-performing scenario, such as with the gratings in Figure 3D), could be attributed to the receptive fields that are formed during the non-associative learning procedure. Specifically, the receptive fields shown seem to all be responsive to a specific orientation and velocity. However, they are all "global" (or "large" scale), in the sense that they all have only one big contiguous area of positive weights along one big contiguous area of negative weights.

We would like to highlight that while our model’s non-associative learning mechanism enables the self-organisation of spatiotemporal receptive fields, the extent to which it can refine patterns or develop multi-scale sensitivity remains an open question. Future work could explore whether large-scale receptive fields in the lobula can be combined or diversified to encode more complex, hierarchical features, thereby improving pattern discrimination.

Regarding the underperformance of the network in certain cases, such as gratings in Figure 3D (Figure 6D in the revised manuscript), we acknowledge that this may be attributed to the receptive fields formed during the non-associative learning procedure. As observed, the receptive fields tend to be global, with contiguous positive and negative weight areas, typically tuned to specific orientations and velocities. This structure may limit the network’s ability to capture finer-scale spatial features, potentially explaining reduced performance in tasks requiring higher spatial resolution. However, our results suggest that bees may approach stimuli for closer inspection and utilise sequential learning strategies when faced with more complex visual tasks, as observed in our previous studies. While this is a prediction of our model, it requires empirical validation in future studies. This aspect is discussed in interpreting the model’s failure to solve the task introduced in Figure 6F.

Additionally, our findings indicate that scanning a restricted image region enhances pattern discriminability, whereas whole-image scanning leads to reduced performance due to weaker and less selective activations in the lobula (Figure 5). This aligns with behavioural evidence suggesting that localised scanning strategies may improve visual encoding in bees, enabling them to extract more informative features from complex stimuli.

We have now expanded the discussion to address these points, emphasising the potential need for receptive field diversity, multi-scale feature encoding, and adaptive scanning strategies as promising future directions for refining the model.

The question that naturally arises is whether the non-associative learning employed here can produce more refined patterns in the receptive fields, or whether it can learn to be sensitive over different scales of features by combining (at the level of the lobula) different large-scale receptive fields in non-trivial ways. Also, why do patterns learned, when scanning whole images, perform best when not applied to a whole image.

Our results show that the model's non-associative learning mechanism enables lobula neurons to develop receptive fields that capture the main statistical features present in natural images, aligning with generative models of efficient coding. Through exposure to diverse natural scenes, the model learns to encode dominant spatial structures such as edges, orientations, and motion patterns, optimising its responses to the most frequently occurring visual features.

While the model successfully extracts key statistical features, the extent to which it can develop more refined receptive fields or achieve multi-scale sensitivity depends on the interactions between non-associative learning and scanning behaviour. In its current form, each lobula neuron integrates visual input over time, forming structured spatiotemporal receptive fields. However, the model has not yet been explicitly trained with different scanning behaviours that might enable it to capture multiple spatial scales simultaneously.

Our findings suggest that non-associative learning effectively captures the dominant statistical features of natural images, but further refinements are needed to enhance multi-scale feature sensitivity. Additionally, the improved performance in localised scanning conditions indicates that targeted feature selection enhances neural encoding, which may reflect an adaptive strategy in biological vision.

The reviewer acknowledges the difficulty in showing the receptive fields when they include both a spatial and temporal component, and as such also that this sensitivity on a varying scale could already be present in the network, as a combination of larger scale receptive fields (albeit the results with different speeds and different distances from the image seems to suggest some specificity in the scale of visual feature that the network can identify). This could warrant a more in-depth study of the performance of the proposed network architecture when varying the training set and training conditions (e.g., speed).

To provide a deeper interpretation of the spatiotemporal receptive field responses, we conducted additional analyses (Figure 3, 5), which have now been incorporated into the main text and discussion in lines relevant to Figures 3 ,5.

5 frames or or..

In this study, we used five fixed frames without explicitly optimizing their order, simplifying the model while aligning with the scanning speed observed in bumblebees. However, this number was tested in control models with different scanning speeds (faster and slower) to evaluate its impact. While we did not determine an optimal sequence, we hypothesize that the number and order of scanning windows feeding input to lobula neurons may vary across individuals and be influenced by their motor dynamics. The interaction between visual sampling and movement is likely an adaptive process, and future studies could investigate whether specific scanning sequences enhance pattern recognition efficiency. Further discussion has been added to the Discussion section to clarify our simplifications and outline future research directions. Additionally, please refer to our responses below for further details.

In-Line commentsL134-136: but is the scanning order selected by the model, or is it fixed? At the moment it seems to be implied that the 5 frames are given. Is there reason to assume this is the optimal order? Is there any optimal order? P.S. I see that this is touched upon below. See comment to L210-219

In this study, for simplicity, we used five fixed frames without explicitly optimising their order. This decision was based on the scanning speed observed in bumblebees during visual exploration. However, while the number of frames remained fixed throughout the experiment, the pixel shift between selected patches was varied in control models with different scanning speeds (faster and slower, Figure 4C) to assess its impact. Although we did not determine an optimal sequence, we hypothesise that the number and order of scanning windows feeding input to lobula neurons may vary among individuals and could be influenced by their motor dynamics. The interaction between visual sampling and movement is likely an adaptive process, and future studies could explore whether specific scanning sequences enhance pattern recognition efficiency (see Discussion).

L185-188: Videos 1 and 2 are missing. I assume they are present in MaBouDi et al. 2021, but if they are referenced here with this indication they should be included. Alternatively, you could add in the text a general reference to the previous paper.

Apologies for the omission. These lines were removed, but the main text still references the videos from MaBouDi et al. (2021).

L196-197: writing here "during the initial experiment" seems to refer to the first experiment you are doing in this paper. Instead, it seems you want to refer to your previous paper. This should be made more clear.

This has now been clarified.

Figure 2B – In the caption, there seem to be typos on what is rewarding and what is punishing.

Corrected.

200-201 – This wording may be interpreted as the model having active control, which is however not the case.

Corrected to sequential scanning.

L201-219: statistical analysis should be done and reported. In an attempt to comparing this model with living animals, I believe every step should be taken to follow the same procedure. How many simulated bees have been tested in the + vs X task? Are they 20 per shape or 20 per scanning pattern? Is the data collected after how many visits? Reporting percentage is I believe insufficient, and a binomial test against chance level should be performed. This is the case for all experiments in this paper.

Thank you for your suggestion. We have conducted various statistical comparisons to evaluate the model’s performance across different experiments and conditions. However, to maintain clarity and focus on the model’s key findings, we have opted to indicate statistical significance in the main text while reporting it in the figures using ‘*’ for p-values less than 0.05 and ‘n.s.’ for non-significant results. Additionally, we have included a dedicated section in the Methods detailing the simulation process (see new section Simulation and Statistical Analysis, in lines 875).

We acknowledge the challenge of directly comparing model outputs with bee behaviour, given the inherent differences between artificial simulations and biological systems. While living organisms exhibit natural variability due to genetic, environmental, and contextual factors, our model operates under controlled conditions with predefined parameters, making direct variability analysis difficult. However, despite these constraints, our model effectively captures key aspects of bee pattern recognition behaviour. Notably, the relative significance of responses—how strongly the model favours one pattern over another—closely aligns with behavioural data, regardless of absolute response variance.

Regarding your specific queries, in the + vs X task, each experimental condition was tested on 20 independent simulations per shape, with randomly initialised neural connectivity to account for individual variability (analogous to testing 20 bees). The data collection procedure follows behavioural study protocols, where each simulated bee undergoes multiple training exposures before performance is assessed. Additionally, each simulation was evaluated with test patterns multiple times, and averages along with the standard error of the mean (SEM) are now reported in the figures to provide a statistically robust representation of the results.

Figure 2B: is this SD or CI?

This has been clarified in the revised version as standard error of the mean (SEM).

L210-219: the authors here refer to an initial poor performance. In figure 2C, is this the last two bars of the graph? This should be made clear.

Apologies for the lack of clarity. This refers to the condition where the entire pattern is scanned. We have now clarified this in the main text.

Overall, the experiments here described aimed at finding the best scanning procedure, but I am not sure if how this was evaluated is appropriate. First, how is the training on natural images organized? Are they all scanned in the bottom half, left to right? If that is the case, the highest efficiency of lower-half scanning may be linked to the highest similarity to training, not to the real efficiency of the technique, and it would as such suggest low generalizability of the model. If instead training is repeated for all experiments following that presentation pattern, and thus the scanning procedure has an effect of learning effectiveness, this may be a property of this network, but not necessarily of living bees (as L216 somewhat suggests). It could in fact be a self-fulfilling prophecy (behavioral experiments suggest the need for scanning, the network is designed with a recurrent layer to enable shape reconstruction, and the network is most effective with scanning).

Regarding training on natural images, the model was not exclusively trained using lower-half scans. Instead, it was trained on full natural images captured at varying distances from natural objects such as flowers, with scanning applied as part of each image's presentation. The natural images were processed sequentially, with patches sampled from different locations over time, simulating the sequential nature of active vision. Importantly, scanning direction and region were not predetermined during training, ensuring that the model was exposed to a broad range of visual structures without inherent bias toward specific scanning trajectories. Given the variability in natural images, the model was effectively trained on a diverse dataset, allowing it to develop a generalised representation of visual patterns.

To further investigate how scanning influences neural representations, we analysed lobula neuron responses to whole-image inputs versus lower-half inputs. Our results show that restricting the input to the lower half led to greater neural discriminability and sparser responses, effectively providing a more selective and efficient signal to higher-level processing in the learning centre. This resulted in improved performance when simulated bees selectively scanned specific regions of the patterns. These findings suggest that localised scanning may enhance information encoding, optimising visual discrimination by reducing redundant inputs.

As discussed in the Discussion section, further studies are needed to determine the optimal scanning strategy that may provide even more efficient coding. Our approach serves as a foundation for future investigations into how dynamic scanning behaviours influence neural encoding and learning in both biological and artificial systems.

It is important to emphasise that our intent was not to assume that scanning is necessary but rather to test whether motion-driven encoding enhances visual discrimination. The model’s architecture does not inherently favour scanning—it is fully capable of processing static images. However, our findings indicate that scanning enhances feature encoding by leveraging spatiotemporal information, allowing for more efficient pattern recognition. This aligns with empirical studies in bees, which actively move to extract visual information rather than passively viewing static stimuli.

In summary, while our results indicate that sequential scanning improves discrimination, we do not claim that the specific trajectory tested represents the optimal strategy for bees. Instead, we propose that motion-driven encoding plays a fundamental role in shaping neural representations, and future studies should explore how different scanning strategies influence visual processing across both biological and artificial systems.

If you want to suggest that scanning from the bottom left is indeed more effective, you need to also include conditions other than the confirmatory one. These could be scanning right to left, or scanning left to right in the top half, or scanning top to bottom, or even diagonally (which I suspect are going to produce identical results). As of now, the experimental conditions only allow us to conclude that scanning sections is more effective than seeing the whole image, which again is to me included as a property of the network. Also, I may be wrong about this, but bees visual field is not centered frontally on the animal, but points upward (https://www.researchgate.net/publication/326717773_Bumblebee_visual_allometry_results_in_locally_improved_resolution_and_globally_improved_sensitivity). Being this the case, a bee moving across the bottom of a stimulus wouldn't it actually be looking at it fully, with the visual field centered on the horizontal symmetry line?

Thank you for raising these points. We acknowledge that our current experiments primarily demonstrate that scanning a specific section of the pattern enhances discrimination compared to viewing the entire image, rather than definitively establishing that scanning from the bottom left is the most effective strategy. To further investigate this, we have expanded our discussion (lines 505–518; 544–558) to emphasize the need for exploring more dynamic scanning trajectories to develop a more realistic model of active vision.

As highlighted in the Discussion section, a bee’s scanning strategy is likely influenced by multiple factors, including task complexity, stimulus properties, and environmental context. However, as the first computational study to examine the role of motion in shaping neural representations of the visual environment, our primary goal is to demonstrate how motion facilitates efficient spatiotemporal encoding. Our findings provide a computationally testable prediction that active scanning plays a fundamental role in structuring neural representations, laying the groundwork for future empirical and theoretical studies.

A similar reasoning should be made for scanning speed. Velocity is tangled with stimulus size. 0.1m/s may work best with this size but will change drastically depending on how much of the stimulus occupies the virtual bee visual field.

Please refer to our response to the previous comment.

I want to point out that none of these points are detrimental to the effectiveness of the model itself, which seems to present good performances. But if claims want to be made about the best scanning strategy, especially if confronted with real animals, these points should be tested and addressed. As of now, we can say that the current model best performs under certain conditions, but we can't generalize the effectiveness of such conditions to be the best for the task, nor the best for bees.

Thank you for your thoughtful feedback. We acknowledge the limitations regarding generalising the scanning strategy as the optimal one for bees. Our primary aim was to investigate how active scanning influences neural representations and pattern recognition rather than to establish a universally optimal scanning strategy.

We agree that determining the best scanning strategy requires a more comprehensive analysis, including testing alternative scanning trajectories and validating these against empirical bee behaviour. While our results demonstrate that the model performs best under specific conditions, we do not claim that these conditions necessarily represent the most effective strategy for bees in real-world settings. Instead, they provide a computationally grounded prediction that scanning behaviours influence neural encoding and recognition performance, which can be tested in future behavioural experiments.

As discussed in the revised manuscript (see Discussion, lines 544-558), scanning strategies in bees are likely influenced by multiple factors, including task complexity, environmental context, and individual variability. Given that bees adapt their behaviour to optimise foraging efficiency and learning, their scanning patterns may not be rigidly fixed but rather dynamically adjusted based on the specific demands of the task. Therefore, further empirical work is needed to explore how bees modulate their scanning strategies in different visual environments and whether certain trajectories lead to more efficient encoding.

L221-252: I believe also here binomial statistics should be produced. I understand it seems to be redundant for performances nearing 100%, but this becomes more relevant for the 60% and 40% reported in Fig3E. On the same note, specific values should be reported, both for averages and SD.

Following your suggestion, we have conducted additional statistical analyses on the model’s output, which are now reported in all result figures using ‘*’ for significance and ‘n.s.’ for non-significant results (see lines 875–893).

246-251 – Repetition of a period.

Corrected.

It would be helpful if Figure 3 also reported the real bees data, as taken from the various papers. This would give a sense of how closely the model follows the bees behavior. Of course, bees are more complex and are subject to, among others, motivational effects which will make the choice percentage less clean, but I still think this would be appreciable.

Directly comparing our model’s performance with all existing bee experimental data is beyond the scope of this study. Instead, we selected specific example studies, each addressing different aspects of pattern recognition challenges in bees. Given that our model represents a simplified visual system, it does not allow for a direct comparison with all behavioural data, particularly since bee responses are context-dependent and vary across experimental conditions.

However, our results suggest that motion plays a beneficial role in visual coding, providing a sparse and selective filtering of visual input before it reaches the learning centre. This structured encoding may facilitate associative processing by enhancing the link between visual patterns and reinforcement signals. While Figure 6 does not include direct behavioural data overlays, we compared them with the real data in the main text.

311-312 – "When the model is limited to only four neurons in the lobula, it lacks the capability to encode the entire spatio-temporal structure that is naturally present in the training patterns". This wording seems to suggest that with more neurons it can encode the entire spatio-temporal structure of the training patterns, which may be an overstatement.

This has been revised and clarified to enhance understanding.

L314-316: I agree that these neurons are sufficient for the discrimination task in hand, but I am unsure whether is appropriate to extend this to bees, as the paragraph title implies. Bees have to use the system to respond to much more complex patterns, like photorealistic ones. For example, is 16 neurons still enough for the face discrimination task?

Determining the upper capacity of bees in pattern recognition is challenging, as defining visual pattern complexity remains an open question in vision science. Moreover, bee responses are highly context-dependent, influenced by environmental factors, prior experience, and task demands. While our model identifies a sufficient number of neurons for the specific discrimination tasks tested, this should not be directly extrapolated to all visual recognition challenges bees encounter in natural settings. Further behavioural neurobiological studies are needed to investigate whether a larger number of neurons is required to encode higher-dimensional visual features and support more complex pattern recognition tasks. However, we have examined this limitation of our model in Figure 6F and further expanded on it in the revised Discussion section.

355-356 – It is unclear how the study would suggest a crucial role of movement in the ability to efficiently analyze and encode the environment. In this work, movement of the input pattern is taken as a given condition under which the network is trained, and not as a tool that can be exploited to have an advantage in the encoding and analysis of the pattern itself.Discussion: In general, I am not fully convinced that your model can say anything about the bees or the optimal performance in general, but should focus on the effectiveness of the model itself. This is because of what I have reported above about how the model performance is at least partially dependent on the model design, and not on how bees actually behave (which is hypothesized)

Our model does not claim to prove that movement is crucial for pattern recognition in bees; rather, it demonstrates how movement-driven spatiotemporal encoding influences neural representation and recognition efficiency.

– Movement is not merely a given condition in our approach; instead, it acts as a structural constraint that shapes encoding strategies in lobula neurons. The model shows that spatiotemporal integration from sequential scanning leads to a sparse and decorrelated representation, which enhances pattern recognition (Figures 2 and 5). Furthermore, experiments comparing different scanning speeds, and the absence of movement (Figure 4) reveal a significant impact on model performance, supporting the importance of motion in efficient visual processing.

– These findings align with experimental observations in bees, where active scanning has been shown to improve visual encoding, as opposed to processing entire images in a single fixation (see introduction). Our results suggest that this scanning strategy may enhance encoding efficiency, reinforcing previous behavioural and neurophysiological evidence.

Regarding the broader biological relevance of our model, we agree that its performance is influenced by our simplification in design choices, and we do not claim that it directly predicts optimal strategies in bees. Instead, our goal is to explore how motion-driven encoding affects visual learning and to generate testable hypotheses for future experimental studies, as discussed in the Discussion section (lines 505-518).

The revised manuscript more clearly distinguishes between model-intrinsic properties and biologically hypothesised mechanisms, ensuring that our claims are well-supported, appropriately framed, and accurately contextualised.

374-375 – In these lines, it is claimed that the model acts as a linear generative model, however, this is not shown in the results and these generative capabilities are not demonstrated.

Thank you for highlighting this point. We have now clarified this claim in the Discussion section (lines 524–535) by explicitly describing the model's generative properties and providing a more detailed explanation of how its spatiotemporal encoding framework captures key statistical features of natural scenes (Figures 2S, 3, 5).

487 – Calling $r_l^{La}$ as "the output of one lamina neuron" instead of "one lamina neuron" could improve clarity.

Corrected.

498 – I have not clear what "however" refers to, in this context

This section has been revised.

498 – Similarly. Rewording "the input of the m −the medulla neuron is calculated" to something like "the input to the $m$-th medulla neuron, $I_m{Me}$, is calculated as"

Corrected.

506-507 – Could reference to Figure 3B be a typo?

Corrected.

528-530 "At each step of training, a set of five patches with size 75x75 pixels, selected by shifting 15 pixels over the image from the left or right or the reverse orientation (Figures1B, 2A), was considered as the input of the model." This wording could be a bit confusing, especially as, coincidentally, 15*5=75. It could be improved to make it clear that one input to the network is (a concatenation?) of 5 patches of 75x75 pixels each, obtained by shifting a window of 75x75 pixels by 15 pixels, 5 times (if this is actually the case).

Thank you for pointing it out. The entire Methods section, particularly this section, has been revised for improved clarity and explanation.